# Rethinking Conditional Diffusion Sampling with Progressive Guidance

**Anh-Dung Dinh**
School of Computer Science
The University of Sydney
dinhanhdung1996@gmail.com

**Daochang Liu**
School of Computer Science
The University of Sydney
daochang.liu@sydney.edu.au

**Chang Xu** *
School of Computer Science
The University of Sydney
c.xu@sydney.edu.au

## Abstract

This paper quantifies and tackles two critical challenges encountered in classifier guidance for diffusion generative models, i.e., the lack of diversity and the presence of adversarial effects. These issues often result in a scarcity of diverse samples or the generation of non-robust features. The underlying cause lies in the mechanism of classifier guidance, where discriminative gradients push samples to be recognized as conditions aggressively. This inadvertently suppresses information with common features among relevant classes, resulting in a limited pool of features with less diversity or the absence of robust features for image construction. We propose a generalized classifier guidance method called Progressive Guidance, which mitigates the problems by allowing relevant classes' gradients to contribute to shared information construction when the image is noisy in early sampling steps. In the later sampling stage, we progressively enhance gradients to refine the details in the image toward the primary condition. This helps to attain a high level of diversity and robustness compared to the vanilla classifier guidance. Experimental results demonstrate that our proposed method further improves image quality while offering a significant level of diversity as well as robust features. Source code is available at: https://github.com/dungdinhanh/prog-guided-diffusion.

## 1 Introduction

Diffusion model [1, 2, 3, 4, 5] has emerged as a state-of-the-art technique for generating high-fidelity images without relying on adversarial training techniques like GANs [6, 7, 8, 9, 10]. This approach has gained popularity among researchers because it avoids mode collapse and training instability. For diffusion models, the generated samples can be further enhanced with improved quality and controllable attributes using guidance techniques for sampling. *Classifier guidance* [11, 12, 13] is one of these techniques which offers numerous benefits. Conditional information can be provided to unconditional diffusion models using noise-aware classifiers' gradients to guide towards pre-defined conditions during sampling. This technique is able to not only improve image quality significantly but also trade-off between diversity and conditional information to fit with the application design. More critically, the flexible nature of classifier guidance allows easy extensions of off-the-shelf diffusion models at the sampling stage without any expensive re-training.

---

*Corresponding authors

37th Conference on Neural Information Processing Systems (NeurIPS 2023).

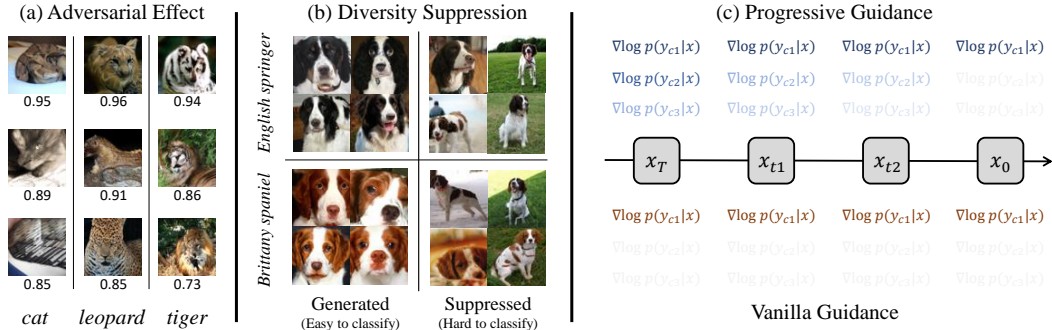

Figure 1: *(a) Samples generated by vanilla classifier guidance achieve high confidence in the condition class but have suspicious features. (b) Images generated by vanilla guidance (left) show a lack of diversity where front-face features are over-exploited. Features shared among breed classes are often lacked, such as action pose and background details depicted in the ground truth images (right). The vanilla guidance has exploited the most distinguishable features to satisfy the classifier while suppressing common features. (c) Our proposed guidance scheme allows gradients from relevant classes to join the sampling process. $c_1$ is the condition. The darkness of the gradients represents the gradients' weights (information degree) during sampling. More examples can be found in Supplementary. Dataset: ImageNet64x64*

In spite of all of its benefits, the classifier guidance still suffers from two critical flaws. The first flaw is the *adversarial effect* where the model fails to produce samples belonging to the target condition. Guiding the model with the gradient and maximizing the probability of the target condition may lead to shortcut learning that generates samples attacking the classifier. Examples can be found in Figure 1(a), where samples achieve very high confidence in the conditional class but result in very poor features. The second flaw is the *diversity suppression* caused by the conflict between the diffusion model's generative nature and the classifier's discriminative nature. The optimization towards the condition overly encourages the model to generate only image features that can be easily recognized by the classifier while neglecting other features relevant to the condition, which can reduce the diversity of the generation. Figure 1(b) shows that common features among classes are suppressed since they are harder to classify, resulting in a lack of diversity. These two problems of the classifier guidance come from its underlying guidance mechanism. Specifically, this approach assumes that the objective of the generative task is the same as that of the discriminative task. By utilizing the gradient $\nabla \log p(y = y_c|x)$, the guidance aims to create an image with features that can be distinguished by the classifier as belonging to the label $y_c$. However, when this approach maximizes the probability of the condition class, it also undesirably suppresses important information relevant but in other classes, causing the lack of diversity and the adversarial effect, as examples mentioned.

This paper proposes a new guidance approach for diffusion models, which generalizes the classifier guidance to tackle the above problems while maintaining its flexibility and efficiency. Our core idea is to alter the aggressive gradients towards a single target in the naive classifier guidance by the gradients that are *progressive* along two dimensions, i.e. the class dimension and the temporal dimension as illustrated in Figure 1(c). Along the class dimension, our method tolerates the gradients from other classes relevant to the condition of the generative task. Along the temporal dimension, we expect the gradients to construct shared features among relevant classes when the image is noisy, and the gradients focus on features more specific to the condition after the image becomes clearer. This scheme allows a larger feature pool, which circumvents information suppression and lack of diversity. Besides, the help from relevant information and the less aggressiveness toward one condition also results in more robust features in the samples.

In summary, our contributions are as follows: (1) Proposing a generalized classifier guidance method with progressive gradients that consider relevant information associated with multiple classes depending on the noise level at each timestep. (2) In-depth analysis on reducing the adversarial effects and the diversity suppression of classifier guidance sampling quantitatively and qualitatively. (3) Improvement on image generation task to reach State-of-the-art (SOTA) on different metrics with different datasets.

## 2    Background

**DDPM**: The Denoising Diffusion Probabilistic Model (DDPM) is represented by the probability distribution function $p_\theta(\mathbf{x}_0) := \int p_\theta(\mathbf{x}_{0:T})d\mathbf{x}_{1:T}$, where $\mathbf{x}_1, \mathbf{x}_2, ..., \mathbf{x}_T$ are latent variables having

the same dimensionality as the data $\mathbf{x}_0 \sim q(\mathbf{x}_0)$. The variable $\mathbf{x}_T$ follows the distribution $p(\mathbf{x}_T) = \mathcal{N}(\mathbf{x}_T; \mathbf{0}, \mathbf{I})$. The process of $p_\theta(\mathbf{x}_{0:T})$ is known as the *reverse process* and is a the Markovian chain: $p_\theta := p(\mathbf{x}_T) \prod_{t=1}^{T} p_\theta(\mathbf{x}_{t-1}|\mathbf{x}_t)$, where $p_\theta(\mathbf{x}_{t-1}|\mathbf{x}_t) := \mathcal{N}(\mathbf{x}_{t-1}; \mu_\theta(x_t, t), \Sigma_\theta(x_t, t))$. This process is also known as the diffusion model's sampling process or inference process.

In contrast to the *reverse process*, the *forward process* corrupts the original data $\mathbf{x}_0$ to $\mathbf{x}_T$ with a predefined schedule of Gaussian noise. This process is a fixed Markovian chain: $q(\mathbf{x}_{1:T}|\mathbf{x}_0) := \prod_{t=1}^{T} q(\mathbf{x}_t|\mathbf{x}_{t-1})$, where $q(\mathbf{x}_t|\mathbf{x}_{t-1}) := \mathcal{N}(\mathbf{x}_t; \sqrt{1-\beta_t}\mathbf{x}_{t-1}, \beta_t\mathbf{I})$. $\beta_t$ is the fixed variance scheduled for both *forward* and *reverse* process.

There are a number of ways to define the output of the network $\theta$ such as noise predictor $\epsilon_\theta(\mathbf{x}_t, t)$, mean predictor $\tilde{\boldsymbol{\mu}}_\theta(\mathbf{x}_t, t)$ [1]. The most common method is the noise predictor $\epsilon_\theta(\mathbf{x}_t, t)$. After the $\theta$ are trained to match $\epsilon_\theta(\mathbf{x}_t, t)$ with $\epsilon$, we have the sampling equation:

$$\mathbf{x}_{t-1} = \frac{1}{\sqrt{\alpha_t}}(\mathbf{x}_t - \frac{1-\alpha_t}{\sqrt{1-\bar{\alpha}_t}}\epsilon_\theta(\mathbf{x}_t, t)) + \sigma_t\mathbf{z}, \tag{1}$$

with $\mathbf{z}$ as the random noise, $\sigma_t$ is the variance of the of $p_\theta(\mathbf{x}_{t-1}|\mathbf{x}_t)$ and $\epsilon_\theta$ is the predicted noise.

**Classifier Guidance**: The guidance aims to provide conditional information during the sampling process so that the output image satisfies the predefined conditions. From Eq. 1, we denote $\mu_t := \frac{1}{\sqrt{\alpha_t}}(\mathbf{x}_t - \frac{1-\alpha_t}{\sqrt{1-\bar{\alpha}_t}}\epsilon_\theta(\mathbf{x}_t, t))$. Given condition $y_c$, $\log p_\phi(y_c|\mathbf{x}_t)$ is the log of conditional distribution of the class $y_c$ through observing $\mathbf{x}_t$. The sampling equation for $\mathbf{x}_{t-1}$ given $\mathbf{x}_t$ with guidance scale $s$ is as below:

$$\mathbf{x}_{t-1} \sim \mathcal{N}(\mu_t + s\sigma_t^2\nabla_{\mathbf{x}_t}\log p_\phi(y_c|\mathbf{x}_t), \sigma_t). \tag{2}$$

Where $\phi$ is the parameters of the pretrained classifier.

## 3  Methodology

We first present a scheme allowing relevant information gradients to join the sampling process. After that, the progressive schedule is proposed to adapt the relevant gradients taking the noise level in samples into account. From the Eq. 2, the short form of sampling with guidance is as below:

$$\mathbf{x}_{t-1} = \mu_t + \sigma_t * \mathbf{z} + s\sigma_t^2\nabla_{\mathbf{x}_t}\log p_\phi(y_c|\mathbf{x}_t). \tag{3}$$

The expected output of $\mathbf{x}_{t-1}$ is to maximize $p_\phi(y_c|\mathbf{x}_t)$. Due to the probability property of $\sum_{i=1}^{C} p(y_i|\mathbf{x}_t) = 1$, the increase in $p(y_c|\mathbf{x}_t)$ will result in the reduction in other probability for other classes $p(y_i|\mathbf{x}_t), \forall i \neq c$ with $c$ is the condition, and $C$ is the total number of classes. This results in the suppression of other classes of information which contributes to harmful effects, as mentioned in Figure 1(a) and (b). We propose to generalize the Eq. 3 as follows:

$$\mathbf{x}_{t-1} = \mu_t + \sigma_t * \mathbf{z} + w\sum_{i=1}^{C} s_i\sigma_t^2\nabla_{\mathbf{x}_t}\log p_\phi(y_i|\mathbf{x}_t), \tag{4}$$

with $s_i \geq 0$ is the degree of information injected into the diffusion sampling process by class $y_i$. We further constraint $\sum s_i = 1$ and $s_c > s_i, \forall i \neq c$. Note that the Eq. 3 is a special case of Eq. 4 where $s_i = 0, \forall i \neq c$ with $c$ is the condition for the sampling process of diffusion models. Similar to [14, 15, 11], a guidance scale $w$ is added to balance denoising signals and classification gradients.

**Remark 3.1** *By incorporating gradients from other classes during sampling, the conflict between the sampling and discriminative objectives can be decreased, resulting in diverse generated samples.*

*Explain.* The use of only the condition class's gradient limits the feature pool's space to some specific features that are the most distinguishable from other classes. This results in the lack of common features shared among relevant classes, which causes a trade-off in the diversity of generated samples. We hypothesize that solely pushing condition gradient suppresses other classes' information since we have $\sum_i p(y_i|\mathbf{x}_t) = 1$ and the increase in $p(y_c|\mathbf{x}_t)$ result in the reduction from others. As a result, the inclusion of other classes' gradients avoids this pitfall.

**Remark 3.2** *Using other classes' gradients beyond the conditional class gradient helps avoid adversarial effects.*

*Explain.* A substantial number of generated images achieve high confidence in the condition class; however, they possess peculiar features unrecognizable to the human eye. Relying solely on conditional class gradients for information construction presents several robustness problems. Firstly, the gradients associated with the conditional class may contain noise that is specific to that class. Depending solely on this information opens a shortcut for the sampling process to exploit the model by utilizing the noise rather than satisfying semantic features. Secondly, suppressing information from other classes has detrimental effects on robust features. Datasets often consist of numerous classes that share common features that play vital role in constructing meaningful images. When these features are suppressed, the image becomes unrecognizable to the human eye.

Using gradients of other classes, in terms of formulation, first helps avoid adversarial attack-like generation through exploiting the sum of gradients $\sum_i \nabla \log p(y_i|\mathbf{x}_t)$. Furthermore, the use of relevant class information forces the model to generate features that satisfy many classes at the same time. This helps avoid specific noise in a class and can utilize the common features shared among classes to avoid the failure of constructing key features. In-depth discussion about the Remark 3.1 and 3.2 are shown in section 4. The rest of this section will discuss the techniques and algorithms to bring the Eq. 4 into the practical sampling process.

### 3.1 Information Degree vector

The information degree $\mathbf{s} = \{s_1, s_2, ..., s_C\}$ is the set of weights associated with the gradients from classes that contribute to the information construction of the image. In order to balance between gradients, we set a constraint $\sum s_i = 1$. Given $c$ is a main condition, $s_c > s_i \ \forall 1 \leq i \leq C$.

From the Eq. 4, for a given condition $c$, the aim is to provide the most relevant features that help avoid information suppression. However, not all labels in the $C$ classes help construct the condition $c$. Instead, for a given class $c$, if the $i^{th}$ class has higher relevance to class $c$ than $j^{th}$ class, we expect to have information degree $s_i > s_j$. There are several ways to satisfy the requirement to have a correlation between labels' features and their degree of information $\mathbf{s} = \{s_1, s_2, ..., s_C\}$, such as Label Enhancement [16] or Partial Multi-label [17, 18]. However, a simple technique, which does not require any training, is preferred due to concerns about the complexity of the whole sampling process. Inspired by the prompt engineering section in CLIP [19], we propose to model the correlation between labels via the description text of labels. Instead of using fixed templates similar to CLIP, we need more details about each class to have more generative features. In this work, we utilize ChatGPT [20] to generate text descriptions for each class. After removing stop words and other preprocessing text, we use a CLIP model to obtain each class's embedding information $v_i$. The embedding information is then utilized to calculate the similarity based on cosine similarity, i.e., $\text{sim}_{i,j} = \frac{v_i * v_j}{||v_i||||v_j||}$. Given the conditional label $c$, we have the information degree for each label as $s_i = \frac{\text{sim}_{c,i}}{\sum_{j=1}^{C} \text{sim}_{c,j}}, \quad \forall 1 \leq i \leq C$. We denote $\mathbf{s}_c = \{s_1^c, s_2^c, ..., s_C^c\}$ as the vector of information degree for sampling the image from condition $c$. To avoid complication in formulation, later in the manuscript, we will only denote $\mathbf{s} = \{s_1, s_2, ..., s_C\}$ for the information degree vector for the condition $c$ without losing generality.

### 3.2 Progressive Information Degree

The subsection 3.1 describes a combination of gradients among classes. In the diffusion sampling, the image $\mathbf{x}_{t-1}$ is less chaotic and more meaningful than the sample $\mathbf{x}_t$. Therefore, we also progressively adapt the $\mathbf{s}$ at every timestep to be less chaotic. We index the $\mathbf{s}$ at each timestep as $\mathbf{s}_t = \{s_{1,t}, s_{2,t}, ..., s_{C,t}\}, \forall 0 \leq t \leq T$. A simple heuristic schedule is proposed to reduce the chaos of $\mathbf{s}$ at each step. At every timestep, for the condition $c$, the degree of information $s_{c,t}$ is increased by a small amount $\Delta s_{c,t}$. At the same time, the $\Delta s_{c,t}$ is distributedly deducted from information degrees $s_{i,t}$ of other classes, ensuring condition $\sum_{i=1}^{C} s_{i,t} = 1$. $\Delta \mathbf{s}_t$ is formulated as:

$$\Delta s_{i,t} = \begin{cases} -\gamma * (1 - s_{i,t}), & \text{if} \quad i = c \\ -\Delta s_c * \frac{s_{i,t}}{\sum_{j=1, j \neq c}^{C} s_{j,t}}, & \text{if} \quad i \neq c \end{cases}, \tag{5}$$

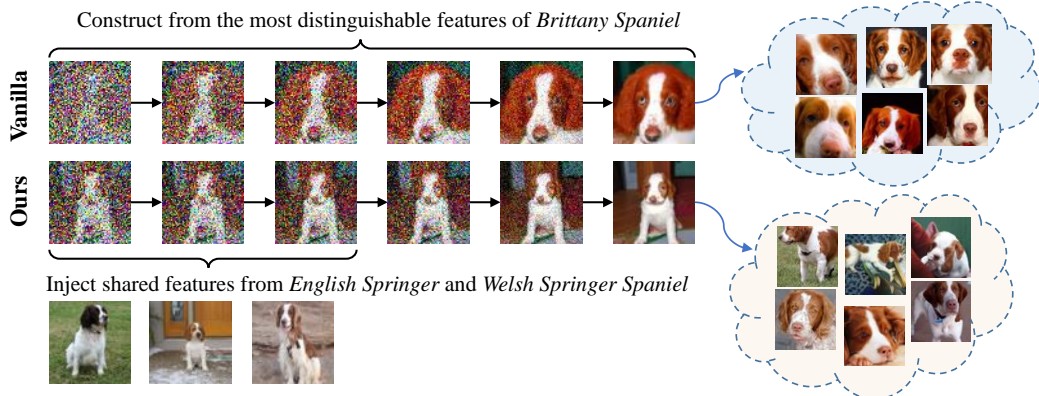

Figure 2: *Diversity Suppression Alleviation. Brittany Spaniel is the condition. Conventional guidance approach often collapses generated samples to primarily exhibit the facial features of Brittany Spaniel. However, by incorporating features from other classes, such as English Springer and Welsh Sprinter Spaniel, the sampling can capture common characteristics shared between classes such as backgrounds, poses and actions.*

with $0 \leq \gamma \leq 1$ as the updating rate. After obtaining $\Delta \mathbf{s}_t$, we update the $\mathbf{s}$ following $\mathbf{s}_{t-1} = \mathbf{s}_t - \Delta \mathbf{s}_t$. From the above equations, if $\gamma = 1$, the sampling process is primarily similar to the vanilla guidance diffusion sampling as in Eq. 2 where only features from class $c$ join in the sampling process.

Next, we use an entropy perspective to interpret the mechanism of our method. With the sampling equation 4, we can generalize the sampling with Kullback–Leibler divergence as:

$$\mathbf{x}_{t-1} = \mu_t + \sigma_t * \mathbf{z} - w\sigma_t^2 \nabla_{\mathbf{x}_t} D_{KL}(\mathbf{s}||p_\phi(\mathbf{y}|\mathbf{x}_t)). \tag{6}$$

When $\mathbf{s}_t$ is fixed, the optimization of $D_{KL}(\mathbf{s}||p_\phi(\mathbf{y}|\mathbf{x}_t))$ totally depends on $\log p_\phi(y_i|\mathbf{x}_t)$. However, when $\mathbf{s}_t$ is varying, the minimization of KL divergence term will take the full form as $D_{KL}(\mathbf{s}_t||p_\phi(\mathbf{y}|\mathbf{x}_t)) = \sum_{i=1}^{C} s_{t,i} \log s_{t,i} - s_{t,i} \log p_\phi(y_i|\mathbf{x}_t)$. Therefore, the process of transforming the vector of information degree can be formulated as below:

$$\min_{\mathbf{s}_t} \quad -\sum_{i=1}^{C} s_{i,t} \log s_{i,t}, \tag{7}$$

so that **(1)** $s_{c,t} > s_{i,t}, \forall i \neq c, 1 \leq i \leq C$, **(2)** $\sum_{i=1}^{C} s_{i,t} = 1$, **(3)** $0 \leq s_{i,t} < 1, \forall 1 \leq i \leq C$, and **(4)** $|s_{i,t}^* - s_{i,t}| \leq l, \forall 1 \leq i \leq C$. With $\mathbf{s}_t^*$ as the optimal solution, we have $\mathbf{s}_{t-1} = \mathbf{s}_t^*$. Because it is hard to optimize the objective 7 and the $D_{KL}(\mathbf{s}_t||p_\phi(\mathbf{y}|\mathbf{x}_t))$ at the same time, the constraint (4) is added with an upperbound $l$ to avoid the mitigation of the minimization of $D_{KL}(\mathbf{s}||p_\phi(\mathbf{y}|\mathbf{x}_t))$ regarding $\mathbf{x}_t$. Using the proposed schedule, the upperbound becomes $l = \gamma * \frac{C-1}{C}$.

Throughout the diffusion model's sampling process, the information degree vector will gradually converge near to one-hot vector, leading to the minimization of objective 7. From this point of view, we have formulated the sampling as the problem of matching two distributions between conditional probability $p(\mathbf{y}|\mathbf{x}_t)$ with information degree vector $\mathbf{s}_t$. The schedule aims to do the reverse entropy regularization on target $\mathbf{s}_t$, which is opposite to the classic entropy regularization [21, 22]. The formal discussion about reverse entropy regularization is in Supplementary.

## 4 Analysis

The previous section has shown our method to guide image generation through a progressive scheme. We now provide further analysis to visualize how our algorithm solves the problems in detail. We set up this analysis on ImageNet64x64 using ADM [11] with or without Progressive Guidance (ProG) to investigate the diversity suppression and the adversarial effect.

**Diversity suppression alleviation**: As the process commences, incorporating information from relevant classes of the conditions into the sampling process is crucial to overcome the issue of information suppression among labels, as illustrated in Figure 2. The Brittany Spaniel is the main condition in this example. We demonstrate that by introducing gradients solely from the Brittany

Spaniel, the process converges towards the features that benefit the pretrained classifier the most, focusing primarily on simple and easily classifiable characteristics — the front face features of the dog. In contrast, by providing information about the English Springer and Welsh Springer Spaniel, we encourage the model to sample images with lower confidence in identifying them as Brittany Spaniels and to tolerate a certain degree of confidence in them belonging to other breeds during the initial stage of the sampling process. This scheme allows the process to search and sample in the pool of common features between classes, facilitating the diverse samples. We further do an analysis of this feature collapse problem on different types of breeds in section B.1 of the Supplementary. We found out that besides *Front-face features* over-exploitation as in Figure 2, the vanilla guidance scheme also suffers from *Front stretching pose* and *Green grass background* problems. Similar to Figure 2, our proposed ProG scheme shows that it can solve these problems.

**Robustness features construction**: Figure 3 provides a visual example of how support from gradients of other classes can effectively reduce non-robust features. In the example, the naive classifier guidance suppresses the representation of *tiger* features, resulting in a significantly degraded image quality due to the lack of facial features of the *leopard*. In contrast, our proposed method promotes the emergence of *tiger*, *panther* and *leopard* features during the early stages of the sampling process. It gradually diminishes *tiger* and *panther* towards the end, ultimately achieving the desired *leopard* features. Additional instances are also demonstrated where our proposed methods effectively address failure cases due to non-robustness features associated with the conventional classifier guidance. More examples with high-resolution images can be found in section C of the Supplementary.

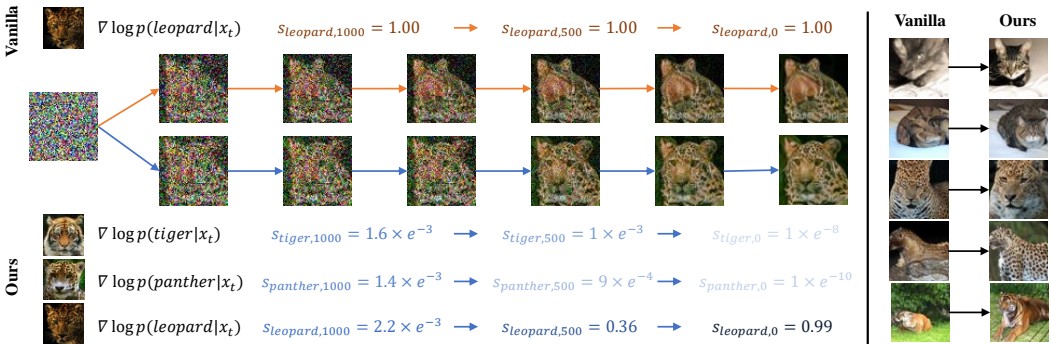

Figure 3: *Robustness Features Construction. The vanilla guidance fails at conditioning the sample as leopard (top-left). The tiger, panther and leopard helps to influence the image content in the initial state to form the critical features of the leopard to overcome the failure (bottom-left), where clear tiger features are observed in the early images. The right part shows more failure cases corrected using our proposed ProG. The darkness of the gradients shows their associated information degree values.*

## 5   Related works

The field of generative models has seen a significant surge in the use of diffusion models. Specifically, the Denoising Diffusion Probabilistic Model (DDPM) and its variants [1, 3, 2, 23, 24, 11, 25] have emerged as prominent models for generative tasks which provides better stability than GANs [26, 27, 28, 29, 30, 9, 31, 32]. Not only for generative tasks, but discriminative tasks also benefit from diffusion scheme [33, 34]. In addition to the DDPM series, score-based models [4, 35] are developed as a theoretical counterpart to the DDPM series. Despite having different optimization objectives and motivations, these two types of models are closely related [5]. In addition to diffusion models, conditional generative models have gained significant attention in the generative model's research community [6, 30]. DGMs [1, 5] offer a conditional version by connecting a classification head to the diffusion model. These models benefit from conditional information [11, 36]. Recent research has focused on achieving controllable samplers/generators without retraining DGMs. Classifier guidance [11] proposes to use classifier gradient signals for guidance, which has been further generalized by [13] to adapt different types of modality. Classifier-free guidance [14] has shown that guidance properties can also be achieved without a classifier. CompDiffusion [37] proposes to guide the diffusion model by combining different conditions given by a pretrained model. However, most guidance works suffer from the trade-off between sample quality, diversity, and conditional

information. [12] has handled this problem by including a scoring model during the training of a noise-aware classifier, causing an increase in the running time. Discriminator Guidance [38] finetunes the sampling process by training a Discriminator, which also faces the problem of large training time similar to [12]. In contrast, our work addresses this problem without retraining the diffusion or noise-aware classifier models. [39] solves the problems of conflict during the sampling process of diffusion models, yet this work ignores the problem of robustness features construction. Adversarial white-box attack [40, 41, 42, 43, 44] shares common technique with classifier guidance to exploit the gradient $\nabla \log p(y_c|X)$ to construct the features, leading to concerns in the community about the use of $\nabla \log p(y_c|X)$ in classifier guidance that causes adversarial effects. One of the main targets of our work is to solve this concern.

## 6 Experiments

**Setup.** Extensive experiments are conducted on CIFAR10, ImageNet (64x64, 128x128, 256x256). We denote Progressive Guidance (ProG) as our proposed method, which is first evaluated on ADM [11] and IDDPM [3] to verify our claims on improving the performance of the vanilla guidance method. Subsequently, ProG is combined with the EDS [45], an advanced guidance method to solve the gradient vanishing problem, to achieve the state-of-the-art. Other baselines are taken from BigGAN[9], ADM [11], EDS[45], IDDPM [3], VAQ-VAE-2[46], LOGAN [28], DCTransformers [47] and DiT[48] (latent diffusion). The ADM with a conditional pretrained diffusion model is denoted as CADM. We utilize five standard scores, which are IS, FID/sFID [26], Precision [49], and Recall [49], to evaluate the image quality and diversity of the generated samples.

Table 1: *ProG helps to achieve better IS/FID/sFID in general. The improvement is significant on both IDDPM [3] and CADM/ADM [11] (both unconditionally or conditionally trained). "-G" postfix attached to diffusion model stands for vanilla classifier guidance. Bold values are the performance our proposed ProG helps achieve better than vanilla guidance. The underlined values are the cases where diffusion models without guidance achieve the best value.* † *is denoted for the score evaluated by the samples provided by the paper.* ‡ *means the values are directly used from the papers due to the unavailability of the pretrained model.*

| MODEL | IS (↑) | FID (↓) | sFID (↓) | PREC (↑) | REC (↑) |
|---|---|---|---|---|---|
| **IMAGENET 64X64** | | | | | |
| ADM | 25.64 | 9.95 | 6.58 | 0.60 | 0.65 |
| ADM-G | 46.90 | 6.40 | 9.67 | 0.65 | 0.54 |
| **ADM-G + PROG** | 46.88 | **5.16** | **6.72** | **0.72** | **0.56** |
| IDDPM | 16.02 | 18.35 | 5.08 | 0.60 | 0.57 |
| IDDPM-G | 18.89 | 13.62 | 4.43 | 0.63 | 0.55 |
| **IDDPM-G + PROG** | **21.60** | **11.12** | **4.25** | **0.67** | **0.55** |
| CADM | 53.79 | 2.07 | 4.35 | 0.73 | 0.63 |
| CADM-G | 66.52 | 2.02 | 4.62 | 0.78 | 0.59 |
| **CADM-G + PROG** | 65.65 | **1.87** | **4.33** | 0.77 | **0.60** |
| **IMAGENET 128X128** | | | | | |
| CADM | 92.53 | 6.14 | 4.96 | 0.69 | 0.65 |
| CADM-G | 141.55 | 2.98 | 5.10 | 0.77 | 0.59 |
| **CADM-G +PROG** | **157.24** | **2.77** | **5.09** | **0.80** | **0.59** |
| **IMAGENET 256X256** | | | | | |
| ADM | 39.7 | 26.21 | 6.35 | 0.61 | 0.63 |
| ADM-G | 96.15 | 11.96 | 10.28 | 0.75 | 0.45 |
| **ADM-G + PROG** | **99.45** | **11.21** | **8.67** | **0.76** | **0.46** |
| CADM | 100.98 | 10.94 | 6.02 | 0.69 | 0.63 |
| CADM-G | 188.91 | 4.58 | 5.21 | 0.81 | 0.52 |
| **CADM-G + PROG** | **222.09** | **4.53** | **5.08** | **0.85** | 0.49 |
| **CIFAR10 32X32** | | | | | |
| ADM | 9.55 | 2.87 | 4.36 | 0.69 | 0.60 |
| ADM-G | 9.58 | 2.85 | 4.30 | 0.68 | 0.60 |
| **ADM-G + PROG** | 9.45 | **2.81** | **4.28** | **0.69** | **0.60** |

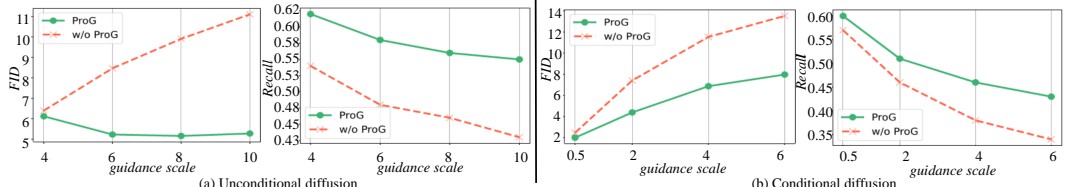

(a) Unconditional diffusion        (b) Conditional diffusion

Figure 4: *FID and Recall trend of guidance sampling with (a) ImageNet64x64 unconditional ADM (b) ImageNet64x64 conditional ADM. For vanilla guidance, it becomes evident that the guidance scale increase leads to a rapid decline in both Recall value and FID, indicating a degradation in the diversity of the samples. Conversely, our method sustains a stable trend associated with $w$ increase.*

**Robustness metric**: The issue of quantifying the adversarial effects caused by the classifier guidance method has not been adequately addressed in previous works [1, 15] since it is challenging due to the trade-off between diversity, feature quality, and robustness. We propose to leverage a bank of off-the-shelf pretrained classification models to evaluate the robustness features based on the assumption that it is much more difficult for the non-robust image to trick a set of separately pretrained classifiers than only a single guidance classifier. Specifically, the robustness features are quantified by averaging Top-1 accuracy using pretrained models ResNet34, ResNet50, ResNet151 [50], DenseNet169, DenseNet201 [51], SqueezeNet, SqueezeNet [52]. To ensure a fair comparison between the two generative schemes, we adjust the guidance scale to achieve similar diversity and feature quality, measured by similarity in the Fréchet Inception Distance (FID).

## 6.1 Improvement over Classifier Guidance

In this section, we show that applying the ProG method helps achieve better performance quantitatively over vanilla guidance regarding image quality, diversity, and robustness. Table 1 shows a clear improvement in all the guidance sampling models on image quality and diversity. ProG achieve lower recall values in some datasets than the ADM baseline without guidance. However, this is expected as conditional generative schemes limit the search space resulting in less freedom in sampling features.

Table 2: *$w$ are adjusted between CADM-G and CADM-G + ProG to achieve similar FID to evaluate robustness. The ProG significantly improves the robustness of the samples.*

| MODEL | ROBUSTNESS ($\uparrow$) | FID |
|---|---|---|
| **IMAGENET 256x256** | | |
| CADM-G | 79.65 | 4.58 |
| **CADM-G + PROG** | **85.04** | **4.56** |

**Diversity trend when increasing** $w$: By increasing guidance scale $w$, the FID scores and diversity scores such as Recall are less sacrificed in ProG to achieve a higher IS score and conditional information compared to vanilla guidance (Figure 4). When increasing the guidance scale, our proposed method mostly has a slower degeneration rate in FID and Recall than the vanilla guidance.

**Robustness improvement** We show that utilizing the ProG method can achieve more robust features than vanilla guidance, with the robustness performance shown in Table 2. We first adjust $w$ so that the two generative processes (vanilla and our proposed) have the same FID scores. After that, we calculate the robustness score as described previously.

**Key takeaway**: The ProG helps to improve the guidance to achieve three targets. Firstly, we achieve better image quality and diversity than the vanilla guidance scheme, which can be observed through IS, FID, sFID, Precision, and Recall. Secondly, we alleviate the diversity suppression qualitatively and quantitatively even when the guidance scale is set to be very large. Finally, we achieved better robustness than the original guidance scheme quantitatively and qualitatively.

## 6.2 State-of-the-Art Comparision

To compare with other state-of-the-art methods, we combine our proposed method with EDS [45]. While our method aims to solve information suppression, the EDS will help amplify the guidance signals at the end of the sampling process. The result shows favorable outcomes in Table 3. Our ProG achieves the state-of-the-art IS/FID on ImageNet64x64 and ImageNet128x128. On ImageNet256x256, the ProG helps to improve DiT-G slightly but consistently over all metrics. Furthermore, we can achieve a comparable level of robustness to classifier-free guidance in Table 4, which is a method

that does not share common techniques with adversarial attacks while also exhibiting a significantly improved diversity trend compared to vanilla classifier guidance, as shown in Table 4. It is worth noting that in the diffusion ADM model, classifier-free guidance outperforms classifier guidance; however, this comes at the cost of nearly twice the computational burden due to the need for forwarding to diffusion twice at each timestep. Moreover, classifier-free guidance lacks the flexibility offered by the classifier guidance method. We also discuss in-depth the benefits of using classifier guidance with ProG compared to classifier-free guidance in section E in Supplementary. Section F in the Supplementary will discuss several techniques to extend ProG into Text-to-Image Generation. The section D in Supplementary will provide the settings for all experiments.

Table 3: *Combining with the EDS [45] to solve the gradient vanishing problem, our proposed method helps achieve state-of-the-art on IMAGENET64x64 and IMAGENET128x128. On latent diffusion models such as DiT, we can achieve slightly better performance to reach SOTA on this dataset.*

| MODEL | IS (↑) | FID (↓) | sFID (↓) | PREC (↑) | REC (↑) |
|---|---|---|---|---|---|
| **IMAGENET 64X64** | | | | | |
| BIGGAN[†] | 44.99 | 4.06 | 3.96 | 0.79 | 0.48 |
| IDDPM* | 46.31 | 2.90 | 3.78 | 0.73 | 0.62 |
| CADM + CLS-FREE* | 63.39 | 1.93 | 4.49 | 0.77 | 0.60 |
| CADM-G + EDS | 61.91 | 1.85 | 4.36 | 0.76 | 0.61 |
| **CADM-G + EDS + PROG** | **65.89** | **1.77** | **4.25** | **0.77** | **0.61** |
| **IMAGENET 128X128** | | | | | |
| BIGGAN[†] | 145.93 | 6.02 | 7.18 | 0.86 | 0.35 |
| LOGAN[‡] | 148.2 | 3.36 | | | |
| CADM-G + EDS | 160.2 | 2.58 | 4.92 | 0.78 | 0.58 |
| **CADM-G + EDS + PROG** | **175.31** | **2.47** | **4.92** | **0.80** | **0.58** |
| **IMAGENET 256X256** | | | | | |
| BIGGAN[†] | 202.77 | 7.03 | 7.29 | 0.87 | 0.27 |
| DCTRANS[‡] | - | 36.51 | 8.24 | 0.36 | 0.67 |
| VQ-VAE-2[‡] | - | 31.11 | 17.38 | 0.36 | 0.57 |
| IDDPM[‡] | - | 12.26 | 5.42 | 0.70 | 0.62 |
| CADM + CLS-FREE | 191.31 | **3.76** | **4.87** | 0.80 | **0.55** |
| CADM-G + EDS | 212.51 | 3.96 | 5.0 | 0.82 | 0.52 |
| **CADM-G + EDS + PROG** | **232.86** | 3.84 | 5.0 | **0.83** | 0.51 |
| DIT (LATENT DIFFUSION) | 122.62 | 9.62 | 6.89 | 0.66 | 0.670 |
| DIT-G | 274.69 | 2.27 | 4.58 | 0.82 | 0.58 |
| **DIT-G + PROG** | **278.77** | **2.25** | **4.56** | **0.82** | **0.58** |

Table 4: *FID and Recall trend for guidance sampling with ImageNet256x256 conditional diffusion. ProG not only helps to achieve better FID but also a higher robustness score. Besides, the diversity trend is also better preserved than vanilla guidance when the guidance scale is increased.*

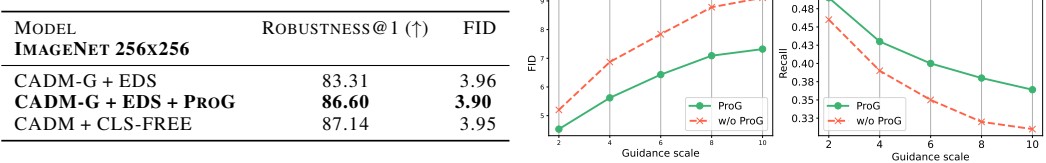

| MODEL | ROBUSTNESS@1 (↑) | FID |
|---|---|---|
| **IMAGENET 256X256** | | |
| CADM-G + EDS | 83.31 | 3.96 |
| **CADM-G + EDS + PROG** | **86.60** | **3.90** |
| CADM + CLS-FREE | 87.14 | 3.95 |

## 6.3 Ablation Study

**Effects of schedule**: ProG schedule relies on a single hyperparameter $\gamma$ to determine the speed of convergence towards the one-hot vector. A larger $\gamma$ leads to faster convergence, while a smaller $\gamma$ results in slower convergence. Table 5 demonstrates the sensitivity of $\gamma$, showing that varying it does not significantly impact the results. Results obtained with $\gamma$ values of $0.04$ or $0.06$ are similar. When $\gamma$ becomes larger, the model is more likely to achieve a higher Inception Score (IS), but trade-off with FID/sFID scores, indicating a decrease in sample quality. It is worth noting that without the schedule ($\gamma = 0$), the guidance method fails, underscoring the importance of the proposed scheduler.

**Correlations between labels**: Fixing the same schedule, we change the initial vector of information degree $\mathbf{s}_T$ via different ways. We denote our ChatGPT-generated text as CGT. Based on CGT, we use CLIP[19] or Word2Vec [53]to obtain the similarity matrix mentioned in section 3. We compare with the uniform label smoothing (Uni. LS), where the $\mathbf{s}_T$ is uniformly distributed. We also provide the performance of Label Smoothing without our progressive schedule noted as LS (NO SCHE.) to have a comprehensive understanding of the method. Table 6 shows that the CLIP + CGT performs the best among different schemes. This shows the importance of the initial state of the $\mathbf{s}_T$, which helps to decide the relevant information for the main condition.

Table 5: $\gamma$ *sensitivity comparision.*

| MODEL IMAGENET 64x64 | IS ($\uparrow$) | FID ($\downarrow$) | sFID ($\downarrow$) |
|---|---|---|---|
| $\gamma = 0.04$ | 46.88 | 5.16 | 6.72 |
| $\gamma = 0.06$ | 48.56 | 5.4 | 7.31 |
| $\gamma = 0.1$ | 56.84 | 7.28 | 9.58 |
| $\gamma = 0.2$ | 57.75 | 8.67 | 12.34 |
| $\gamma = 0.0$ (NO SCHE.) | 7.97 | 73.10 | 19.77 |

Table 6: *Different approaches for* $\mathbf{s}_T$ Eq.6

| MODEL IMAGENET 64x64 | IS ($\uparrow$) | FID ($\downarrow$) | sFID ($\downarrow$) |
|---|---|---|---|
| CLIP + CGT | 46.88 | **5.16** | **6.72** |
| W2V + CGT | **55.69** | 6.72 | 8.79 |
| UNI. LS | 26.66 | 16.39 | 13.13 |
| LS (NO SCHE.) | 25.32 | 9.98 | 6.62 |

**Computational cost**: Considering the comparable FID/sFID results between ProG and classifier-free guidance, we uncovered several advantages of ProG over classifier-free guidance, as detailed in Section E of the Supplementary. Notably, one of these advantages pertains to computational efficiency. Table 7 illustrates the GPU hours required for generating 50,000 images at a resolution of 256x256. When ProG is incorporated into the standard classifier guidance framework, the computational cost remains on par with vanilla classifier guidance. In contrast, uti-

Table 7: Computational cost to generate 50000 images with ImageNet256x256

| MODEL IMAGENET256x256 | GPU HOURS |
|---|---|
| ADM | 236 |
| CADM-G + EDS | 341 |
| **CADM-G + EDS + PROG** | 341 |
| CADM + CLS-FREE | 487 |

lizing classifier-free guidance substantially escalates computational expenses. It's worth noting that the computational cost of classifier guidance is predominantly influenced by the architecture of the noise-aware classifier, which is typically simpler than the network architectures used in generative models. Consequently, the observations in Table 7 are broadly applicable in most scenarios.

## 7 Conclusion

This work quantifies two problems of the classifier guidance method: dramatic diversity suppression and non-robustness feature construction. The results show better robustness features than the classifier guidance baseline and a similar level to the classifier-free guidance method. Besides, we also achieve a significantly better diversity trend when increasing the guidance scale quantitatively and qualitatively. We can achieve the SOTA on ImageNet64x64 and ImageNet128x128 on FID/sFID. Our method can be combined with DiT-G [48] to achieve a new SOTA for ImageNet256x256. Our current research demonstrates successful solutions to address the adversarial effects, particularly in cases where high-confidence generated samples exhibit minimal features relevant to the given condition. However, we acknowledge that we have not yet resolved the scenario where diffusion signals dominate and conflict with the classification signals. The resulting images have low confidence and fail to contain any discernible features related to the specified condition. It is worth noting that these features can no longer be considered adversarial due to low confidence and often be wrongly classified by the pretrained classifier used in the sampling process. Resolving the conflict between diffusion and classification gradients remains an open challenge and an area for future investigation.

## Acknowledgement

This work was supported in part by the Australian Research Council under Projects DP210101859 and FT230100549. Besides, the AI training platform supporting this work was provided by High-Flyer AI (Hangzhou High-Flyer AI Fundamental Research Co., Ltd.)

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
