# 1 Supplementary

## 2 A Details of sampling process

3 The sampling process with ProG is shown below. The correctness is shown in the section I. All hyperparameters are in Table 14.

---
**Algorithm 1** DDPM denoising process with ProG guidance

---
**Input:** class labels $y$, classification scale $s$
$\mathbf{x}_T \sim \mathcal{N}(\mathbf{0}, \mathbf{I})$
$\mathbf{s}_T \leftarrow$ initialized via semantic correlations in section 3.1
**for** $t = T, ..., 1$ **do**
$\quad z \sim \mathcal{N}(\mathbf{0}, \mathbf{I}), g \leftarrow w \sum_{i}^{C} s_{i,t} \nabla_{\mathbf{x}_t} \log p_\phi(y|\mathbf{x}_t)$
$\quad \mathbf{x}_{t-1} \leftarrow \frac{1}{\sqrt{\alpha_t}}(\mathbf{x}_t - \frac{1-\alpha_t}{\sqrt{1-\bar{\alpha}_t}}\epsilon_\theta(\mathbf{x}_t, t)) + \sigma_t^2 g + \sigma_t z$
$\quad \Delta s_{i,t} = \begin{cases} -\gamma * (1 - s_{i,t}), & \text{if} \quad i = c \\ -\Delta s_c * \frac{s_{i,t}}{\sum_{j=1, j \neq c}^{C} s_{j,t}}, & \text{if} \quad i \neq c \end{cases}$,
$\quad s_{i,t-1} \leftarrow s_{i,t} - \Delta s_{i,t}, \forall 1 \leq i \leq C$
**end for**

---

## 4 5 B Diversity analysis

### 6 B.1 Qualitative analysis

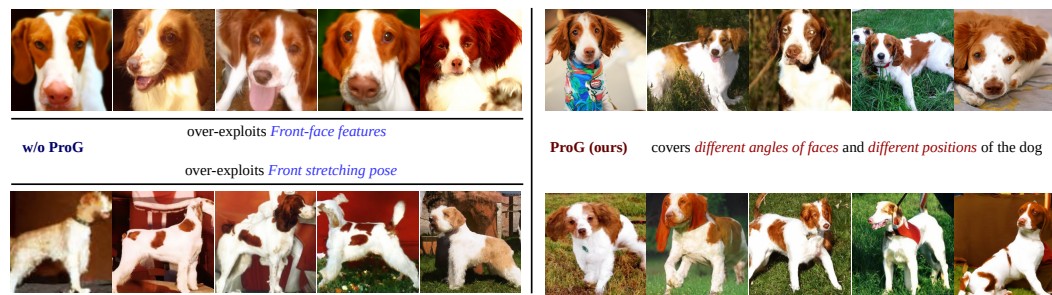

w/o ProG — over-exploits *Front-face features* / over-exploits *Front stretching pose*

ProG (ours) — covers *different angles of faces* and *different positions* of the dog

Figure 5: *Brittany Spaniel condition. The left figure shows the first and the second type of diversity suppression of vanilla classifier guidance where only front-face features and front-stretching pose features are exploited. The right figure is the diversity improvement using our proposed ProG. ProG can helps to cover a wide range of features in the class.*

7 This section will extend the analysis of Remark 3.1 and section 4. More examples and patterns of
8 avoiding a lack of diversity are presented. All the analyses are visualized on ImageNet 256x256 with
9 three breeds *Brittany Spaniel*, *English Springer* and *Welsh Springer Spaniel*.

10 As stated in the main paper, the absence of diversity stems from suppressing shared features among
11 classes. These features are challenging to classify because multiple classes possess them, resulting
12 in diminished significance in discriminative tasks like classification. This leads to the ignorance
13 of common features. The main paper's Figure 2 and 1(b) provide visual insights into this notion.
14 Consequently, we have discovered that various classes exhibit distinct patterns of diversity suppression
15 based on their shared feature pool with others. Through observations across different breeds, we have
16 identified three primary instances of feature collapse resulting from the suppression of other features:

17      1. Collapse into front-face features

18      2. Collapse into a single pose

19      3. Collapse into one type of background.

20 Most of the breeds will have the first type of collapse as in Figure 5, 6, and 7. Figure 5, 6 shows the
21 improvement over front-face features collapse and single pose collapse using ProG. Figure 5 and 7
22 shows the improvement of ProG on front-face collapse and background collapse.

Figure 7 shows the $1^{st}$ and $3^{rd}$ types of collapse into front-face features and one type of background.

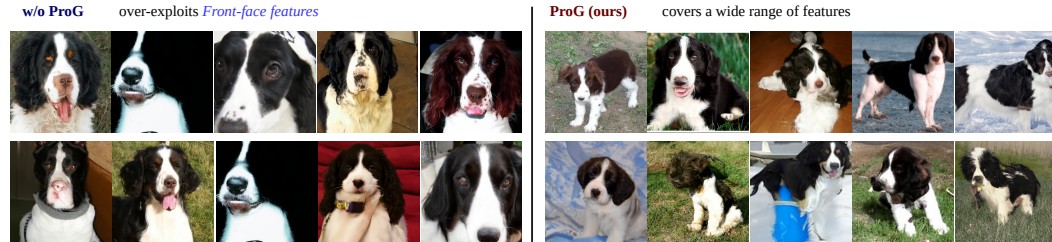

Figure 6: *English Springer condition. The left figure shows a clear collapse into front-face features by using vanilla classifier guidance. The right figure shows our improvement where different poses, backgrounds and angles of faces are recovered.*

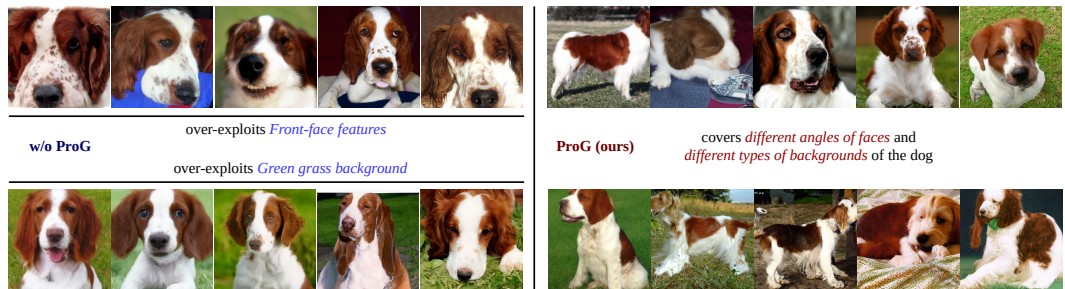

Figure 7: *Welsh Springer Spaniel. (left-figure) Similar to other breeds, the vanilla guidance also over-exploits the front-face features, and it also over-exploited the green grass background features. The right figure shows the improvement using ProG, where different backgrounds are recovered.*

23

24 **Background correction**: We also provide more evidences on improving the background collapse
25 case by our proposed method. We analyze on the class *Briard*, where the background collapse
26 happens very strongly. Figure 8 shows the correction case-by-case using ProG.

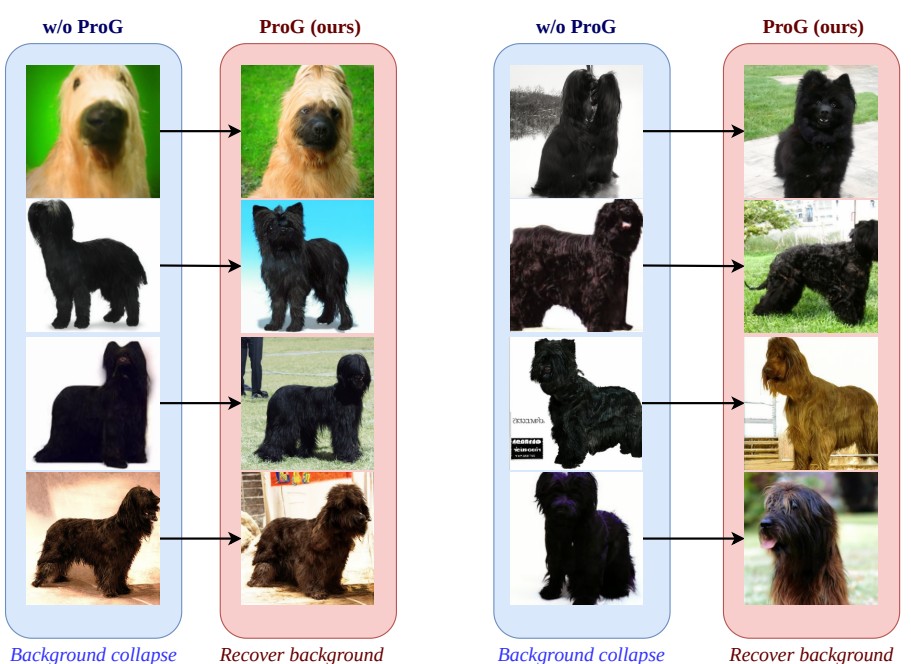

Figure 8: *Briard condition. On vanilla guidance (w/o ProG), most of the images fall into white/green simple background. We leverage ProG to improve the background as well as the details of the dog.*

## B.2  Quantitative analysis

We also extend the results from section 6.1 at Figure 4 and the two figures in Table 4 in main paper.

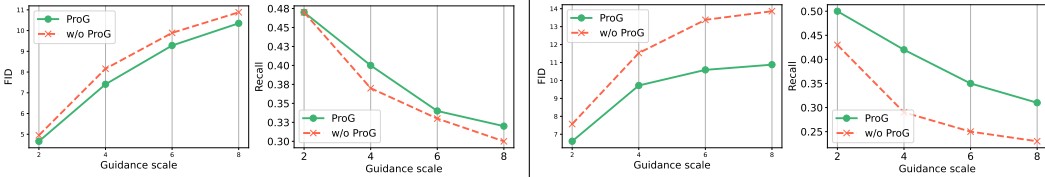

Figure 9: *FID and Recall trend of guidance sampling with (left) ImageNet128x128 conditional ADM (right) ImageNet128x128 conditional ADM with EDS. Opposite to the degeneration of diversity in vanilla guidance, our method sustains a stable trend associated with $w$ increase.*

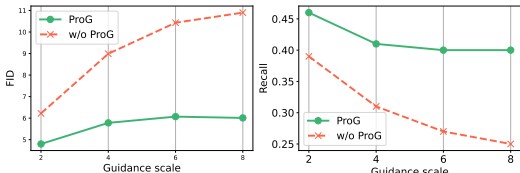

Figure 10: *FID and Recall trend of guidance sampling with ImageNet 256x256 with conditional diffusion models. We see a clear improvement in trend over vanilla guidance when increasing $w$*

Figure 9 and 10 show a clear improvement in keeping diversity when increasing guidance scale $w$. Given a very large weight for classifier gradients, it is not necessary to trade off the diversity to achieve clearer examples following the main condition.

## C  Robustness analysis

We also show that by avoiding pushing toward one condition so aggressively, we can avoid non-robust features that can be used to attack the classifier models. The non-robust feature images are the images with very high confidence to belong to a class but have suspicious features toward that class.

In Figure 1(a) (main paper), a number of images with very high confidence belong to the condition but have very poor features. Figure 11 shows that those cases can be overcome by using the proposed ProG. We further evaluate the robustness feature constructions on ImageNet 256x256. Examples from this dataset are given in Figure 12 as the *Brittany Spaniel* class, 13 as *Briard* class and 14 as the *Leopard* class. The values under the images are the confidence of that image belonging to a class. This confidence is measured by the classifier used for guidance.

The results are consistent with the robustness score recored in Table 2 and 4 in the main paper.

## D  Experimental details

This section will provide information about the settings of every experiment in the main paper and the Appendix. All the hyperparameters are shown in Table 14. $\gamma$ is selected as the best value by running on two values, $0.04$ and $0.06$. $w$ is selected based on similar scheme of the paper [1].

### D.1  Settings

All the experiments are executed on HPC clusters with 8 A100-40GB GPUs with Ubuntu 20.04 operating system. The total RAM for each node is 400GB.

**Robustness metric**: The issue of quantifying the adversarial effects caused by the classifier guidance method has not been adequately addressed in previous works [2, 3] since it is challenging due to the trade-off between diversity, feature quality, and robustness. We propose to leverage a bank of off-the-shelf pretrained classification models to evaluate the robustness features based on the assumption that it is much more difficult for the non-robust image to trick a set of separately pretrained classifiers than only a single guidance classifier. Specifically, the robustness features are quantified by averaging Top-1

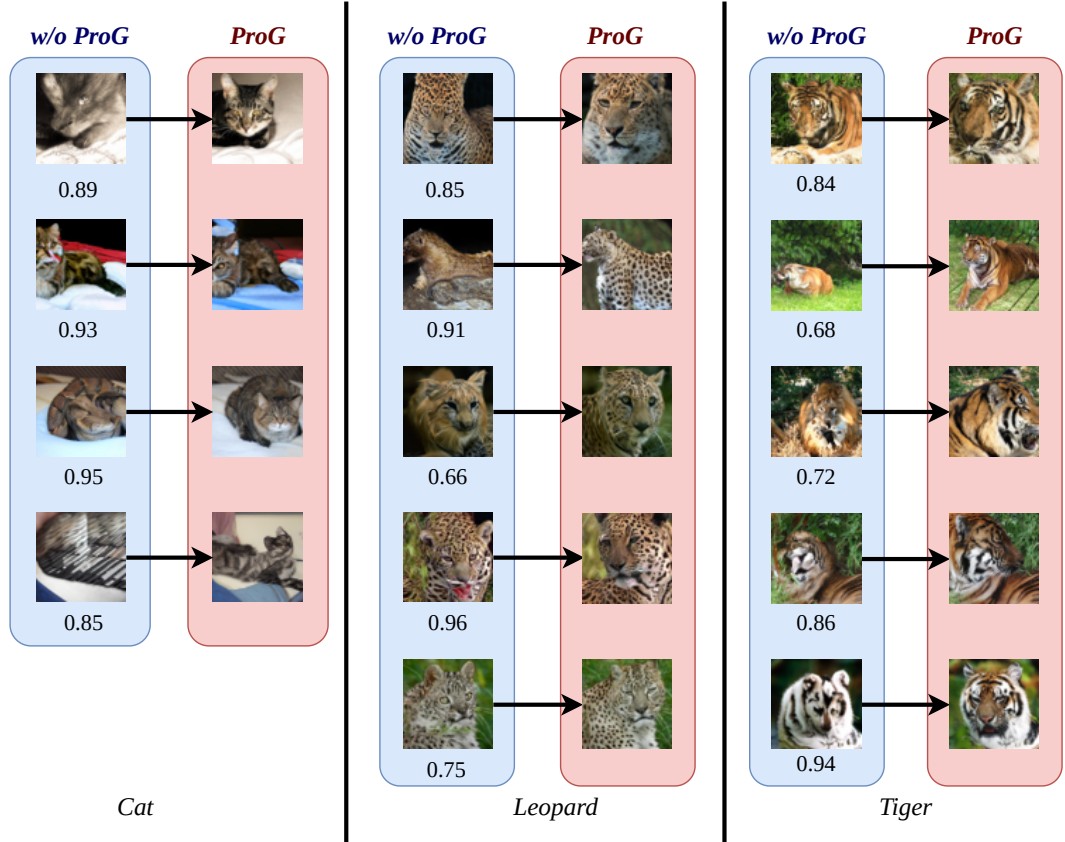

Figure 11: *We show the non-robustness avoidance using our proposed ProG compared to the examples show in Figure 1(a) in the main paper.*

accuracy using pretrained models ResNet34, ResNet50, ResNet151 [4], DenseNet169, DenseNet201 [5], SqueezeNet, SqueezeNet [6]. To ensure a fair comparison between the two generative schemes, it is crucial to establish a comparable level of quality and diversity in the synthetic data generated by both models. This is essential because if one model generates inferior features compared to the others but repetitively produces only a few easily distinguishable features across the entire dataset, it could achieve a higher robustness score than the others. To address this, we adjust the guidance scale to attain similar diversity and feature quality between the two schemes, as assessed by the Fréchet Inception Distance (FID) metric. The selection of FID is based on its ability to strike a balance between sample diversity and sample quality, whereas other metrics like Recall tend to bias towards diversity and place less emphasis on image quality.

Due to the pretrained ResNet34, ResNet50, ResNet151, DenseNet169, DenseNet201, and SqueezeNet only having available pretrained models on ImageNet 224x224, it is more reliable to evaluate the robustness on synthetic ImageNet 256x256. For other resolutions, we might need to retrain the off-the-shelf classifiers resulting in uncertainty in hyper-parameters and the training process.

### D.2 Details setup for each experiment in section 6

**Classifier guidance improvement**: Section 6.1 shows improved classifier guidance using ProG. All the pretrained diffusion and noise-aware classification models are taken from `https://github.com/openai/guided-diffusion/blob/main/README.md`. CADM is the conditional diffusion ADM. We cannot obtain results for ADM-G and ADM-G + ProG on ImageNet128x128 due to the unavailability of the unconditional diffusion model on this resolution (not provided by the guided-diffusion GitHub folder). Table 1 shows the superiority over original classifier guidance on **image generation** task, and Table 2 shows the better **robustness** score compared to conventional classifier guidance. Figure 4 shows better **diversity** trend when increasing classifier guidance scale $w$.

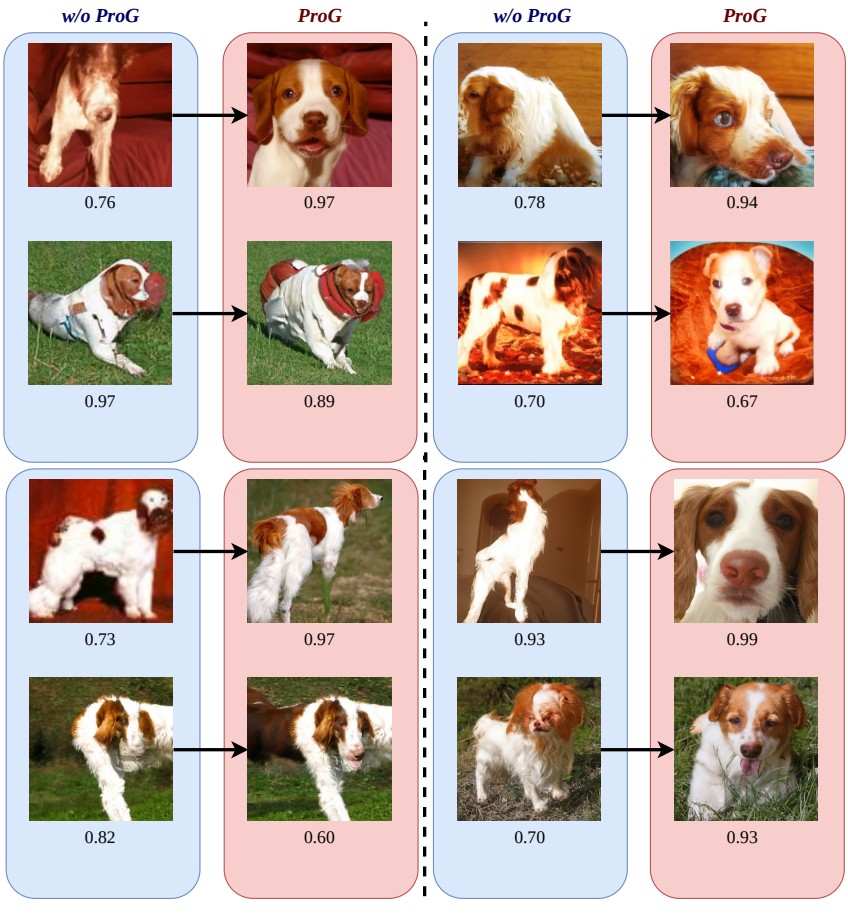

|  |  |  |  |
| :---: | :---: | :---: | :---: |
| *w/o ProG* | *ProG* | *w/o ProG* | *ProG* |

*Brittany Spaniel*

Figure 12: *Brittany Spaniel condition. Vanilla classifier guidance (blue pads) often achieves high confidence regarding the conditions but obtains weird features. Using ProG can help to avoid the non-robust features which result in much more realistic images.*

**SOTA comparison**: Section 6.2 shows the utilization of ProG to achieve SOTA on image generation task. We have the following models BigGAN[7], ADM [1], EDS[8], IDDPM [9], VAQ-VAE-2[10], LOGAN [11], DCTransformers [12] and DiT[13] are basline models. Without any notation, the pretrained model taken from the main paper is utilized for sampling synthetic data for evaluation. † is denoted for the score evaluated by the samples provided by the paper. ‡ means the values are directly used from the papers due to the unavailability of the pretrained model.

CADM+CLS-FREE (**classifier-free guidance**) [14] is sampled by using separate unconditional and conditional diffusion models due to the lack of the pretrained model from the main paper [14]. This approach is reasonable since the authors in [14] verify that classifier-free guidance can work on separate models.

DiT [13] is the conditional **latent diffusion model**, and DiT-G is the classifier-free version of DiT (this is the default setup [13]). To apply ProG on this model, we use a noise-aware classifier on latent space (keep the same architecture as Classifier ImageNet64x64 [1], only replacing the input layer to feed the latent input).

We are aware that the concurrent work by Kim et al. [15] (Refining Generative Process with Discriminator Guidance in Score-based Diffusion Models), currently under review at ICML2023, which presents state-of-the-art (SOTA) results on ImageNet 256x256. However, it is essential to note that their approach achieves these results by training a discriminator model specifically tailored to one diffusion model. While this approach yields impressive outcomes, it significantly limits the flexibility of diffusion sampling. In their proposed scheme, three distinct models must be available

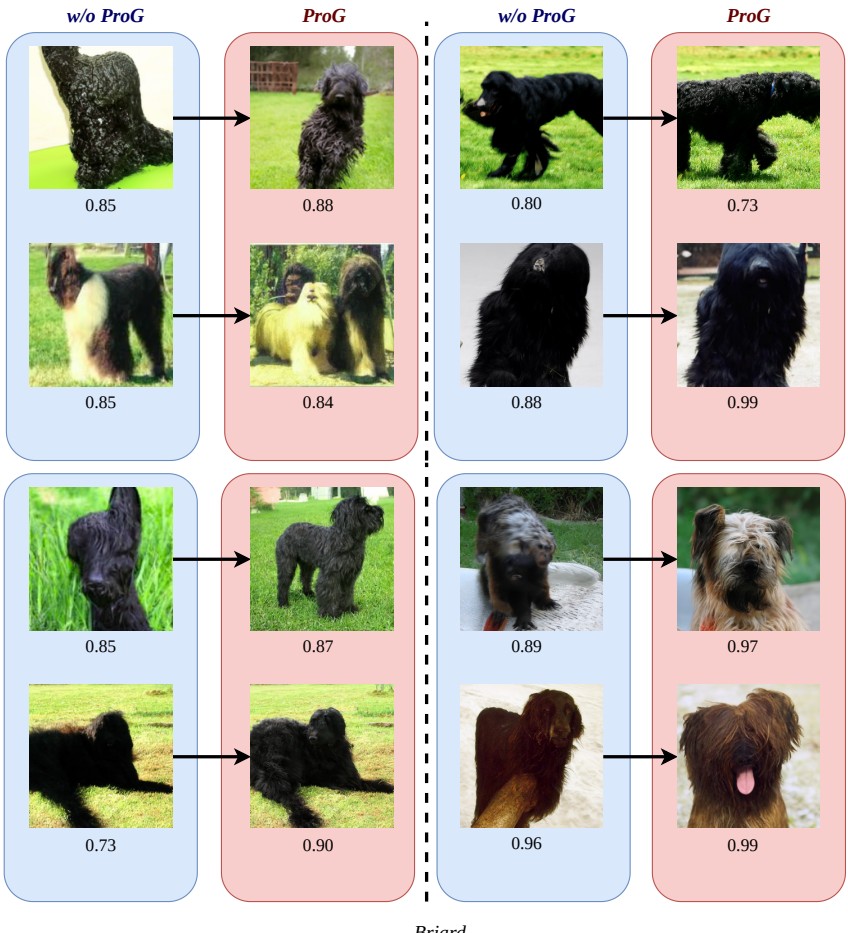

| w/o ProG | ProG | w/o ProG | ProG |

*Briard*

Figure 13: *Briard condition. Vanilla classifier guidance (blue pads) often achieves high confidence regarding the conditions but obtains weird features. Using ProG can help to avoid the non-robust features which result in much more realistic images.*

simultaneously: the diffusion model, the noise-aware classifier, and the noise-aware discriminator. In contrast, our proposed scheme is a plug-and-play scheme with high flexibility.

Moreover, the training process for their discriminator model necessitates the generation of synthetic datasets, adding another computational challenge. It is worth mentioning that this approach allows for a more significant amount of training data compared to both conventional guidance diffusion and our proposed ProG scheme. Consequently, a direct comparison between our proposed ProG and the work by Kim et al. [15] would be inherently unfair due to these substantial disparities in methodology and resource requirements.

Furthermore, it is worth considering that our ProG model can potentially enhance the results achieved by the scheme proposed by Kim et al. [15] in certain scenarios. Given that our models effectively enhance both the robustness features and diversity of synthetic datasets, there exists an opportunity to combine the strengths of our ProG model with the algorithm presented in [15] to address the disparity between synthetic and real data. However, we leave exploring this combination for future work.

# E   Classifier-free discussion

Although the classifier-free guidance suffers from high-cost computation and inflexibility due to the need for both unconditional and conditional diffusion models simultaneously, this model has gained popularity due to concerns about adversarial effects resulting from the shared techniques between classifier guidance and adversarial attacks.

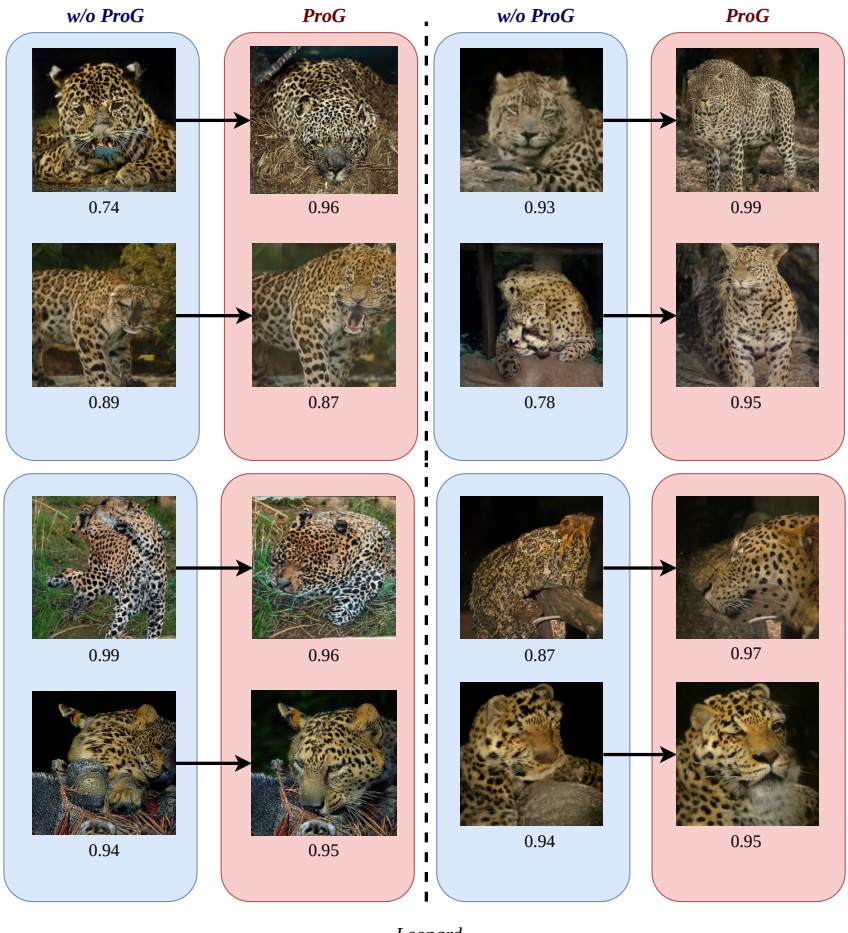

*w/o ProG*     *ProG*     *w/o ProG*     *ProG*

*Leopard*

Figure 14: *Leopard condition. Vanilla classifier guidance (blue pads) often achieves high confidence regarding the conditions but obtains weird features. Using ProG can help to avoid the non-robust features which result in much more realistic images.*

In this section, we demonstrate how ProG addresses these challenges by improving classifier guidance, allowing it to achieve a similar level of robustness as classifier-free guidance while significantly reducing computational costs and achieving a high level of flexibility compared to classifier-free guidance.

Regarding the robustness level, ProG can achieve a similar level of robustness compared to classifier-free guidance, while classifier-free guidance does not exploit the common technique with adversarial attacks as in Table 7.

Table 7: *Robustness score comparison between classifier-free guidance and ProG*

| MODEL IMAGENET 256x256 | ROBUSTNESS(↑) | FID |
|---|---|---|
| **CADM-G + EDS + PROG** | 86.60 | 3.90 |
| CADM + CLS-FREE | 87.14 | 3.95 |

Table 8: *GPU hours to sample 50000 images between ProG and classifier-free guidance*

| MODEL IMAGENET 256x256 | GPU HOURS |
|---|---|
| **CADM-G + EDS + PROG** | 341 |
| CADM + CLS-FREE | 487 |

Regarding diversity, we show that we slightly achieve better diversity compared to classifier-free guidance as in Figure 15 and significantly better in diversity trend when increasing the guidance scale as Figure 15.

In terms of flexibility, we find out that ProG or classifier guidance has several benefits over classifier-free guidance.

**ProG (ours)**

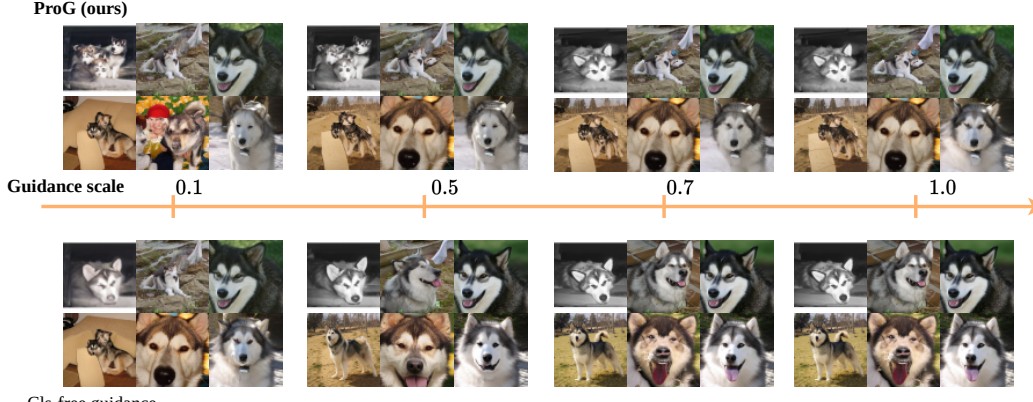

Guidance scale    0.1          0.5          0.7          1.0

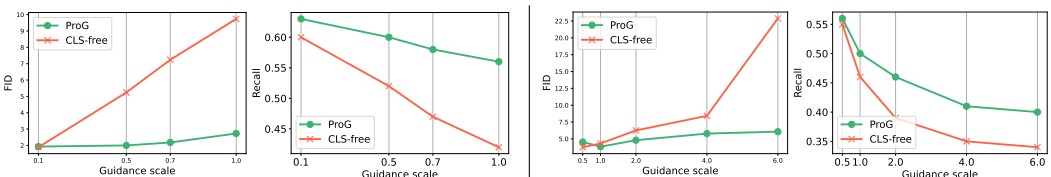

Cls-free guidance

Figure 15: *When increasing the guidance scale, the classifier-free guidance freezes all the dogs to open their mouth gradually, showing the collapse into one style. Our proposed methods modify the generated figures so that it is more realistic without collapsing*

Figure 16: *FID and Recall trends when increasing guidance scale (left) for ImageNet 64x64 and (right) for ImageNet 256x256. ProG achieves better diversity trend when increasing guidance scale compared to classifier-free guidance*

Table 9: *Comparison between the most advanced classifier guidance and classifier-free guidance*

| MODEL | IS (↑) | FID (↓) | sFID (↓) | PREC (↑) | REC (↑) |
|---|---|---|---|---|---|
| **IMAGENET 64X64** | | | | | |
| CADM + CLS-FREE* | 63.39 | 1.93 | 4.49 | 0.77 | 0.60 |
| **CADM-G + EDS + PROG** | **65.89** | **1.77** | **4.25** | **0.77** | **0.61** |
| **IMAGENET 256X256** | | | | | |
| CADM + CLS-FREE | 191.31 | **3.76** | **4.87** | 0.80 | **0.55** |
| **CADM + EDS + PROG** | **232.86** | 3.84 | 5.0 | **0.83** | 0.51 |
| DIT-G | 274.69 | 2.27 | 4.58 | 0.82 | 0.58 |
| **DIT-G + PROG** | **278.77** | **2.25** | **4.56** | **0.82** | **0.58** |

- *Training flexibility*: Different from classifier-free guidance, noise-aware classifier, used in classifier guidance, can be trained separately from diffusion model. This offers the training flexibility for conditions. Whenever the conditions are provided or modified, it is not necessary to re-train diffusion model. Instead, the noise-aware classifier can be fine-tuned or retrained at a much cheaper cost compared to fine-tuning the diffusion models.

- *Sampling flexibility*: Classifier-free guidance can only work when we have both conditional diffusion model and unconditional diffusion model, or at least a joint training between the two models. This reduces the flexibility of the guidance technique when given different versions of the diffusion model with updated training images or different training schemes. In contrast, given the same latent space with diffusion models, the noise-aware classifier imposes no restrictions on the diffusion model. It can be applied to any diffusion model, either trained to be conditional or unconditional or even a combination between condition and null condition.

| | Training flex. | Sampling flex. | Low cost | Robust | Diversity | Extend. |
|---|:---:|:---:|:---:|:---:|:---:|:---:|
| Vanilla guidance | ✓ | ✓ | ✓ | ✗ | ✗ | ✓ |
| CLS-free guidance | ✗ | ✗ | ✗ | ✓ | ✗ | ✓ |
| **ProG** | ✓ | ✓ | ✓ | ✓ | ✓ | ✓ |

Table 10: As we can see, the main reason for the popularity of classifier-free guidance is its robust features. However, ProG can combine all the advantages of Vanilla and Classifier-free guidance in one unified scheme.

142    • *Extendibility*: Both the guidance techniques can be extended to various conditions e.g.
143      Text-to-image.

144  We summarize the benefits of our ProG compared to classifier-free guidance as below:

145  **Main takeaway:** Classifier guidance can outperform classifier-free guidance on low-resolution
146  datasets (ImageNet 64x64) but achieves poorer performance on ImageNet 256x256 on FID. However,
147  the ProG helps the classifier guidance to achieve a similar level of robustness and better diversity
148  compared to classifier-free guidance. It is important to note that classifier guidance offers much more
149  flexible guidance with a very lightweight cost than its classifier-free counterpart (lower GPU hours
150  than classifier-free guidance as in Table 8).

151  # F   Extension to Text2Image Generation

152  Text2Image Generation tasks recently attracted a number of research around the work. As a result,
153  this section extends ProG to work on the Text2Image problem.

154  In general, [1] has proposed to extend classifier guidance for text-to-image guidance. The sampling
155  equation for GLIDE is shown below:

$$x_{t-1} = \mu_t + \sigma_t * \mathbf{z} + s\sigma_t^2 \nabla_{x_t}(f(x_t).g(c))$$

156  Where $f(x_t)$ is the image embedding vector and $g(c)$ is the text or description embedding vector.
157  Equation (1) is mostly similar to equation (3) in the main paper, the only difference is the gradient
158  term resulted from the similarity between two embedding vectors instead of the classification gradient
159  as in the main paper.

160  Our proposed ProG is applied to GLIDE in equation (1) in the following two scenarios:

   • Given one caption, we will utilize a set of 1000, 5000, or 10000 captions to act as relevant
     information to input during the sampling process. we have:

$$g(c) = \sum_{i=0}^{i=N+1} s_i g(c_i)$$

   with $i = 0$ is the index of the primary caption, and $i \neq 0$ is the index of other captions. We
   set the initial values of $s_i$ as:

$$s_i = \frac{g(c_0).g(c_i)}{\sum_{j=0}^{N+1} g(c_0).g(c_j)}$$

161  The value of $s_i$ is progressively updated throughout the sampling process as in section 3.2
162  in the main paper. This scheme is named as **GLIDE-ProG**

   • Given one caption, we use 4 other captions that have the same meaning as the original
     caption but different words. Since four other captions all have the same meaning, we have
     different strategies to set the $s_i$ values:

$$s_i = \begin{cases} a, & \text{if } i = 0 \\ \frac{a}{4}, & \text{otherwise} \end{cases}$$

163  , with $a$ is hyperparameter, we try with $a = 0.3$. This method is named as **GLIDE-ProGsim**

|  | zero-shot FID ($\downarrow$) | GPU hours |
|---|---|---|
| GLIDE | 24.80 | 34.27 |
| GLIDE-ProG w N=1k (ours) | 23.47 | 34.66 |
| GLIDE-ProG w N=5k (ours) | 23.50 | 34.83 |
| GLIDE-ProG w N=10k (ours) | 23.31 | 34.83 |
| GLIDE-ProGsim(ours) | 23.87 | 34.84 |

Table 11: The improvement is significant in all the scenarios (around **6%**), with N is the number of additional captions we used. Dataset: MSCoco64x64

|  | **zero-shot FID** ($\downarrow$) | **GPU hours** |
|---|---|---|
| GLIDE | 34.80 | 38.45 |
| GLIDE-ProG w N=1k (ours) | 32.55 | 45.50 |
| GLIDE-ProG w N=5k (ours) | 32.37 | 45.80 |
| GLIDE-ProG w N=10k (ours) | 32.28 | 46.10 |
| GLIDE-ProGsim(ours) | 31.91 | 46.23 |

Table 12: The improvement is significant in all the scenarios (around **8%**), with N is the number of additional captions we used. Dataset: MSCoco256x256.

We set up the evaluation precisely the same as GLIDE [1] to evaluate zero-shot FID on MS-CoCo. Note: 4 additional captions of GLIDE-ProGsim are taken from MS-Coco captions. 1k, 5k, and 10k captions are randomly sampled from the MSCoco training set. Table 1 and Table 2 shows the evaluation results:

**Main takeaway**: From Table 11 and Table 12, the ProG scheme helps significantly improve the performance of text-to-image guidance on different scenarios.

# G    Classifier performance sensitivity

The classifier's performance during sampling is one of the important indicators of the content in the image. In this section, we explore the correlation between classifier performance sensitivity regarding $\gamma$ in Algorithm 1. As we can observe in Table 13, FID is very sensitive with $\gamma$, which means the generated image quality is heavily affected by $\gamma$. However, the classifier performance at different noise levels has little sensitivity regarding $\gamma$ and has little correlation with the image quality.

# H    ChatGPT prompt discussion

Since ChatGPT is used for replacing human efforts in collecting data, we do not focus much on investigating the performance affected by different ChatGPT prompts. We believe that the research related to the prompts to achieve better performance could result in a more complicated work and leave for the future work. In this paper, we use the prompt that has the format:

*Add text description. For example, "Tench" will turn into "Characterized by its distinctive olive-green to golden-brown coloration, the tench has a robust and slightly elongated body with a rounded tail fin. It inhabits slow-moving or still waters such as lakes, ponds, and slow rivers across Europe*

| $\gamma$ | FID | Acc@25 | Acc@75 | Acc@150 | Acc@200 | Acc@250 |
|---|---|---|---|---|---|---|
| 0.04 | 5.16 | 00.00 | 0.31 | 20.00 | 78.42 | 100 |
| 0.06 | 5.4 | 00.00 | 0.31 | 20.31 | 79.06 | 100 |
| 0.1 | 7.28 | 00.00 | 0.31 | 21.87 | 78.75 | 99.68 |
| 0.2 | 8.67 | 00.00 | 0.31 | 20.62 | 79.37 | 100 |

Table 13: Sensitivity of $\gamma$ regarding FID and the noise-aware classifier accuracy. Acc@25 means the classifier's accuracy at the $25^{th}$ timestep.

 *and parts of Asia. Renowned for its adaptability to varying water conditions, the tench can thrive*
*in environments with low oxygen levels due to its unique respiratory adaptations.". Apply for the*
*following fields:*

- *Goldfish, Carassius auratus*

- *Great white shark, white shark, man-eater, man-eating shark, Carharodon Zacharias*

The primary motivation is hinting at the type of description we want. Suppose we use different types
of prompts that do not have a hint. The output is a lengthy paragraph that includes information
unrelated to the description, such as its origination place or history, which is harder to do text
preprocessing and less relevant to generate image features.

# I   Formulations discussion

The optimization problem, as in Eq. 7, is fully shown as below:

$$\min_{\mathbf{s}_t} \quad -\sum_{i=1}^{C} s_{t,i} \log s_{t,i} \tag{8}$$

$$\text{s.t} \quad s_{c,t} > s_{i,t}, \quad \forall i \neq c, 1 \leq i \leq C \tag{9}$$

$$\sum_{i=1}^{C} s_{i,t} = 1, \tag{10}$$

$$0 \leq s_{i,,t} < 1, \quad \forall 1 \leq i \leq C \tag{11}$$

$$\mathbf{s}_t \in \arg\min_{\mathbf{s}_t} D_{KL}(\mathbf{s}_t || p_\phi(\mathbf{y}|\mathbf{x}_t)) \tag{12}$$

Since the simultaneous optimization of Eq. 8 and Eq. 12 is difficult, we reduce the problem as:

$$\min_{\mathbf{s}_t} \quad -\sum_{i=1}^{C} s_{i,t} \log s_{i,t} \tag{13}$$

$$\text{s.t} \quad s_{c,t} > s_{i,t}, \quad \forall i \neq c, 1 \leq i \leq C \tag{14}$$

$$\sum_{i=1}^{C} s_{i,t} = 1, \tag{15}$$

$$0 \leq s_{i,t} < 1, \quad \forall 1 \leq i \leq C \tag{16}$$

$$|s_{i,t}^* - s_{i,t}| \leq l, \quad \forall 1 \leq i \leq C \tag{17}$$

The Eq. 17 is utilized to replace the Eq.12 with the assumption that $D_{KL}(\mathbf{s}_t || p_\phi(\mathbf{y}|\mathbf{x}_t)) \sim 0$ after $\mathbf{x}_t$
is optimized via Eq. 6 as copy as below:

$$\mathbf{x}_{t-1} = \mu_t + \sigma_t * \mathbf{z} - w\sigma_t^2 \nabla_{\mathbf{x}_t} D_{KL}(\mathbf{s}_t || p_\phi(\mathbf{y}|\mathbf{x}_t)). \tag{18}$$

**Reverse entropy regularization**: As we can see, the problem now turns into the distribution matching
between $p_\phi(\mathbf{y}|\mathbf{x}_t)$ and $\mathbf{s}_t$ at each time step. Since we initialize $\mathbf{s}_T$ very chaotic when the sampling
process starts, it is not difficult for updating $\mathbf{x}_t$ to match with the distribution of $\mathbf{s}_T$. We regularize the
$\mathbf{s}_t$ at each timestep to avoid overfitting similar to [16, 17]. Although there are many possible ways
to move the $\mathbf{s}_t$ ahead, we have to satisfy the condition of $s_{c,t} > s_{i,t}$ at every timestep. Otherwise,
the condition of the generation process can not be reached. As a result, we move the $\mathbf{s}_t$ toward the
direction that reduces the entropy leading to the reverse entropy regularization problem.

**Correctness of the algorithm**: Our proposed schedule ProG can satisfy as a solution to the optimiza-
tion problem Eq. 8. We have:

$$\Delta s_{i,t} = \begin{cases} -\gamma * (1 - s_{i,t}), & \text{if} \quad i = c \\ -\Delta s_{c,t} * \frac{s_{i,t}}{\sum_{j=1, j \neq c}^{C} s_{j,t}}, & \text{if} \quad i \neq c \end{cases}, \tag{19}$$

208  and $\mathbf{s}_{t-1} = \mathbf{s}_t - \Delta\mathbf{s}_t$.

209  Since $s_{c,t}$ monotonically increase every timestep and $s_{i,t}$ monotonically reduces every timestep, we
210  always satisfy $s_{c,t} > s_{i,t} \forall 0 \le i \le c$.

211  After the update $\mathbf{s}_t$ to achieve $\mathbf{s}_{t-1}$, we have:

$$\sum_{i=1}^{C} s_{i,t-1} = s_{c,t-1} + \sum_{i=1,i\neq c}^{C} s_{i,t-1} \tag{20}$$

$$= s_{c,t} + \gamma(1 - s_{c,t}) + \sum_{i=1,i\neq c}^{C} s_{i,t} - \gamma * (1 - s_{c,t})\frac{s_{i,t}}{\sum_{j=1,j\neq c}^{C} s_{j,t}} \tag{21}$$

$$= s_{c,t} + \gamma(1 - s_{c,t}) - \sum_{i=1,i\neq c}^{C} \gamma * (1 - s_{c,t})\frac{s_{i,t}}{\sum_{j=1,j\neq c}^{C} s_{j,t}} + \sum_{i=1,i\neq c}^{C} s_{i,t} \tag{22}$$

$$= s_{c,t} + \gamma(1 - s_{c,t}) - \gamma * (1 - s_{c,t}) + \sum_{j=1,j\neq c}^{C} s_{j,t} = \sum_{i=1}^{C} s_{i,t} \tag{23}$$

212  Given the $\mathbf{s}_T$ is initialized following information degree as in section 3.1 in the main paper, the
213  $\sum_{i=1}^{C} s_{i,T} = 1$ leads to the $\sum_{i=1}^{C} s_{i,t} = 1, \forall t$ satisfying the constraint Eq. 15. Since the deduction
214  to the $s_{i,t}$ follows the distribution of $s_{i,t} \forall 1 \le i \le C, i \le c$, we have $s_{i,t} \ge 0 \forall i, t$ satisfying all the
215  constraints in the problem in Eq. 8.

216  Since the $s_{c,t}$ will move to 1, other $s_{i,t}$ will also gradually move to 0. The process will reduce the
217  entropy as Eq. 8 close to 0, satisfying the entropy minimization.

218  As $\sum_{i=1}^{C} s_{i,t} = 1, \forall t$ and $s_{c,t} > s_{i,t}, \forall i \neq c$, we have $s_{c,t} > \frac{1}{C}$. Upper bound $l = \gamma * (1 - \frac{1}{C}) = \gamma * \frac{C-1}{C}$

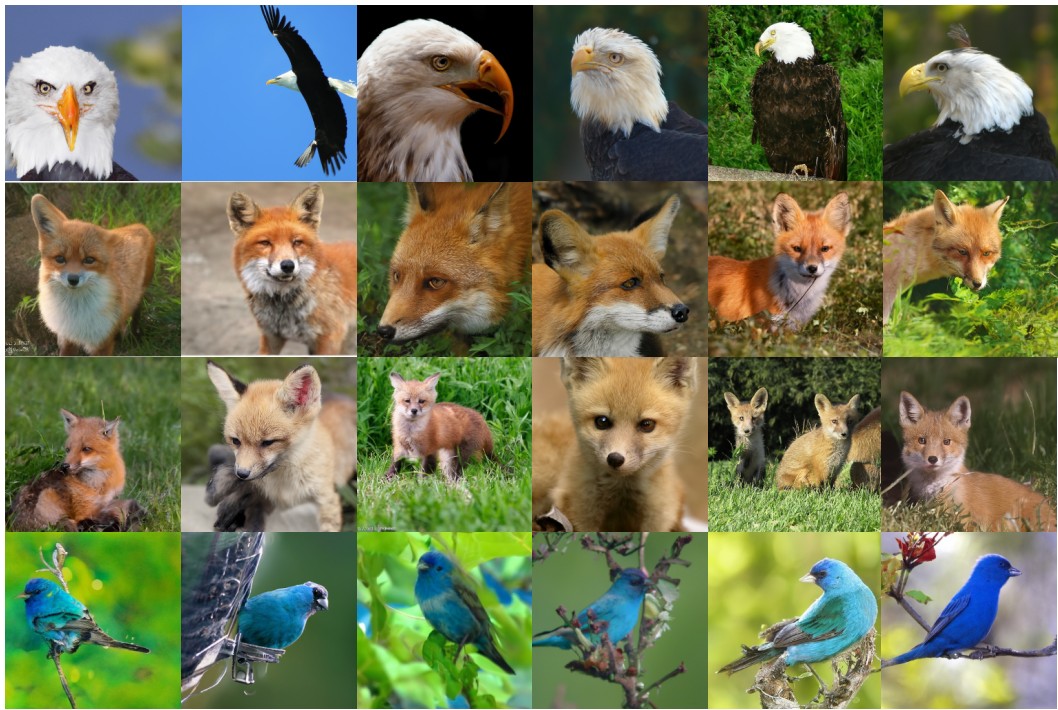

Figure 17: *Images generated by classifier guidance with ProG*

219

Table 14: All hyper-parameters required for reproducing the results.

| MODEL | DATASET | $\gamma$ | $w$ | TIME-STEPS |
|---|---|---|---|---|
| **TABLE 1** | | | | |
| ADM, IDDPM | IMAGENET 64x64, 128x128 256x256 | 0.0 | 0.0 | 250 |
| ADM, IDDPM | CIFAR 32x32 | 0.0 | 0.0 | 250 |
| CADM | IMAGENET 64x64, 128x128, 256x256 | 0.0 | 0.0 | 250 |
| ADM-G | IMAGENET 64x64 | 0.0 | 4.0 | 250 |
| ADM-G + PROG | IMAGENET 64x64 | 0.04 | 8.0 | 250 |
| IDDPM-G | IMAGENET 64x64 | 0.0 | 2.0 | 250 |
| IDDPM-G + PROG | IMAGENET64x64 | 0.06 | 4.0 | 250 |
| CADM-G | IMAGENET64x64 | 0.0 | 0.5 | 250 |
| CADM-G + PROG | IMAGENET 64x64 | 0.06 | 0.5 | 250 |
| CADM-G | IMAGENET128x128 | 0.0 | 0.5 | 250 |
| CADM-G + PROG | IMAGENET 128x128 | 0.06 | 0.7 | 250 |
| ADM-G | IMAGENET 256x256 | 0.0 | 10.0 | 250 |
| ADM-G + PROG | IMAGENET 256x256 | 0.06 | 14.0 | 250 |
| CADM-G | IMAGENET 256x256 | 0.0 | 1.0 | 250 |
| CADM-G + PROG | IMAGENET 256x256 | 0.04 | 2.0 | 250 |
| ADM-G | CIFAR 32x32 | 0.0 | 0.3 | 250 |
| ADM-G + PROG | CIFAR 32x32 | 0.04 | 0.3 | 250 |
| **TABLE 2** | | | | |
| CADM-G | IMAGENET 256x256 | 0.0 | 1.0 | 250 |
| CADM-G + PROG | IMAGENET 256x256 | 0.04 | 2.3 | 250 |
| **TABLE 3** | | | | |
| CADM-G + EDS | IMAGENET64x64 | 0.0 | 0.2 | 250 |
| CADM-G + EDS + PROG | IMAGENET64x64 | 0.04 | 0.2 | 250 |
| CADM-G + EDS | IMAGENET128x128 | 0.0 | 0.5 | 250 |
| CADM-G + EDS + PROG | IMAGENET128x128 | 0.06 | 0.5 | 250 |
| CADM-G + EDS | IMAGENET 256x256 | 0.0 | 1.0 | 250 |
| CADM-G + EDS + PROG | IMAGENET256x256 | 0.04 | 1.0 | 250 |
| DIT | IMAGENET256x256 | 0.0 | 0.0 | 250 |
| DIT-G | IMAGENET256x256 | 0.0 | 1.5 | 250 |
| DIT-G + PROG | IMAGENET256x256 | 0.03 | 1.5 | 250 |
| **TABLE 4 (INC. FIGURES)** | | | | |
| CADM-G + EDS | IMAGENET 256x256 | 0.0 | $1.0 \sim 10.0$ | 250 |
| CADM-G + EDS + PROG | IMAGENET256x256 | 0.06 | $1.0 \sim 10.0$ | 250 |
| **FIGURE 2** | | | | |
| ADM-G | IMAGENET 64x64 | 0.0 | 10.0 | 250 |
| ADM-G + PROG | IMAGENET 64x64 | 0.06 | 10.0 | 250 |
| **FIGURE 3 (SELECT LOW $\gamma$ FOR BETTER OBSERVATION)** | | | | |
| ADM-G | IMAGENET 64x64 | 0.0 | 10.0 | 250 |
| ADM-G + PROG | IMAGENET 64x64 | 0.001 | 10.0 | 250 |
| **FIGURE 5, 6, 7 AND 8** | | | | |
| ADM-G | IMAGENET 256x256 | 0.0 | 14.0 | 250 |
| ADM-G + PROG | IMAGENET 256x256 | 0.04 | 14.0 | 250 |
| **FIGURE 12, 13 AND 14** | | | | |
| ADM-G | IMAGENET 256x256 | 0.0 | 10.0 | 250 |
| ADM-G +PROG | IMAGENET 256x256 | 0.04 | 10.0 | 250 |
| **FIGURE 11** | | | | |
| ADM-G | IMAGENET 64x64 | 0.0 | 10.0 | 250 |
| ADM-G +PROG | IMAGENET 64x64 | 0.04 | 10.0 | 250 |