# OpenReview forum: "Rethinking Conditional Diffusion Sampling with Progressive Guidance"
_NeurIPS.cc/2023/Conference — NeurIPS 2023 poster_

### Official Review · Reviewer_mBBP · 2023-07-01

**Soundness:** 3 good
**Presentation:** 3 good
**Contribution:** 4 excellent
**Rating:** 7
**Confidence:** 5

**Summary:**

This paper proposes a generalized classifier guidance method for diffusion models with progressive guidance along both the class and temporal dimensions, to handle the adversarial effect and diversity suppression problems of the vanilla classifier guidance. In experiments, the proposed method shows advantages over the vanilla classifier guidance and achieves a new state-of-the-art when combined with other methods under certain settings.

**Strengths:**

1. This paper has clear and well-established motivations, working on two important problems in classifier guidance encountered by the community, the adversarial effect and diversity suppression. This can be a good contribution to the community.

2. The proposed progressive guidance method is simple, effective, intuitive, and well-aligned with the motivations. The entropy perspective is also very interesting.

3. The paper is well-written, with good clarity, nice flow, and good intuition. At the same time, detailed explanations and illustrative analysis are provided for the method for better understanding. The reading experience is good.

4. The experiments are relatively comprehensive with multiple datasets and baselines. The alleviation of the adversarial effect and diversity suppression is validated. The advantage over the vanilla classifier guidance is shown. New SOTA is achieved under certain settings.

5. Code is available.


**Weaknesses:**

1. In Table 1, the results on CIFAR are not as good as on ImageNet. What is the potential reason and any insight?

2. The information degree is based on the description generated by ChatGPT. What are the prompts? What is the impact of different prompts?

3. Although the paper is well-written in general, there are still some clarity issues:
- The derivation and introduction of the entropy perspective in the main paper are not very easy to understand. It should be improved in a more logical way in writing.
- What do the superscripts on methods in Table 3 mean? What does “LS (NO SCHE.)” mean in Table 6 mean?

4. There are also some minor writing issues:
- In Figure 1c, the meaning of darkness is not stated.
- In Figure 1c, it should be stated that “c1” is the condition.
- In Figure 1 caption, missing space before “Dataset: ImageNet64x64”.
- In L57, “Propose” > “Proposing”.
- In L61, “over” > “of”.
- In L61, “SOTA” should be full.
- “Markov chain” > “Markovian chain”.
- In section 2, the definition of epsilon and sigma is not mentioned. The background should be self-contained.
- In L79, “log_phi p” should be “log p_phi”. Phi is not explicitly defined.
- In L87-88, p should be p_phi. Make it consistent.
- In L91, >= should be a single symbol.
- The flow at the start of section 3.1 is too fast. Add some rationale and definition of information degree first.
- In L131, missing space after “ChatGPT”.
- In Eq.7, s_t,i should be s_i,t. Also check other places.
- In Table 2, should “CADM-G+ProG” be “CADM+ProG”?
- In L260, remove spacing after “EDS [32]”.
- In Table 3 caption, “State” > “state”.
- The capitalization in the reference should be corrected, such as “gans”.


**Questions:**

Please address the writing issues then I would consider this paper as a good contribution to the community.

**Limitations:**

The limitation is briefly discussed at the end of the paper. I think the authors should elaborate on this to provide more insights.

---

> ### Author Rebuttal · Authors · 2023-08-09
>
>
> ### Question 1: Low improvement on CIFAR-10
> We have analyzed this problem and discovered that the problem lies in the semantic labels of CIFAR10 itself. Compared to ImageNet, the labels in CIFAR-10 has less relevant information than each other. Most of the shared information between classes is background information. For example, an airplane and a bird only share blue background; an automobile might only share a green background or a street background with a horse. As a result, most of the supporting information from other classes quickly turns into noise during the sampling. In contrast, in ImageNet, many classes share similar features, such as a set of breeds, Brittany Spaniel, Standard poodle, keeshond, and Eskimo dog, that share a lot of backgrounds, colors, and poses. The shared information from relevant classes is beneficial for constructing the primary class.
>
> ### Question 2: Prompts and its effect
>
> The prompt I use has the form:
> ```
> Add text description. For example, "Tench" will turn into "Characterized by its distinctive olive-green to golden-brown coloration, the tench has a robust and slightly elongated body with a rounded tail fin. It inhabits slow-moving or still waters such as lakes, ponds, and slow rivers across Europe and parts of Asia. Renowned for its adaptability to varying water conditions, the tench can thrive in environments with low oxygen levels due to its unique respiratory adaptations.". Apply for the following fields:
>
> 1. Goldfish, Carassius auratus
> 2. Great white shark, white shark, man-eater, man-eating shark, Carharodon Zacharias
> ```
>
> The primary motivation is hinting at the type of description we want. Suppose we use different types of prompts that do not have a hint. The output is a lengthy paragraph that includes information unrelated to the description, such as its origination place or history, which is harder to do text preprocessing and less relevant to generate image features.
>
> ### Question 3.1: Introduction and derivation of reverse entropy regularization
> We will re-write this part in the final manuscripts. In detail, we will add an explanation for each constraint as below:
> 1. Constraints (1), (2), (3) are to satisfy the information degree as a distribution with $s_c$ achieves the highest value. This is matched with the conditions of information degree mentioned in section 3.1.
> 2. As mentioned in eq.6, when $\mathbf{s}_t$ is varied,
>
> $$D_{KL}(\mathbf{s} _t || p _{\phi}(\mathbf{y}|\mathbf{x}_t)) = \sum _{i=1}^C s _{t, i} \log(s _{t, i}) - s _{t,i} \log p _{ \phi}(y_i | \mathbf{x} _t)$$
> is considered in full form instead of only $\sum _{i=1}^C -s _{t,i} \log p _{ \phi }(y_i | \mathbf{x} _t)$ as equ. (4).
>
> However due to the term $\sum_{i=1}^C s_{t,i} \log s_{t,i}$, the objective of  $\min _{\mathbf{s} _t} D _{KL}$ and equ. (7) in the main paper is conflict. To avoid this case, we introduce a constraint
>
> $$|s^*_{t,i} - s_{t,i}|  \leq l, \forall 1 \leq i \leq C \quad \quad _{(4)}$$
>
> This constraint (4) helps to keep every value of $s_{i,t}$ can not change larger than the bound $l$ at one timestep, resulting in a minimal change toward KL divergence while minimizing the entropy objective.
>
> ### Question 3.2: Clarification on Table 3 and Table 6.
>
> As we mentioned in lines 83 - 85 Appendix, $+$ is denoted for the score evaluated by the samples provided by the paper. $++$ means the values are directly used from the main article due to the unavailability of the pretrained model. The $*$ over classifier-free guidance means that we don't have available implementation and the pretrained model from the paper. As a result, we do our implementation as well as evaluation. We will state these clearly in the final manuscript.
>
> In Table 6, the LS (NO SCHE.) means we apply Label smoothing for information degree without a progressive schedule. This means that the values of information degree do not change during the sampling. This helps to verify that our improvement does not come from smoothing the gradient scales.
>
> ### Question 4: Minor writing issues
> Since we can not upload the revision, we will edit the writing according to the reviewer's suggestion in the final manuscripts:
> 1. The darkness in Figure 1c means the weight emphasized for that gradient. Darker means more weight put on that gradient.
> 2. In Table 2, "CADM-G + ProG" is utilized consistently throughout the paper. Thus, it is correct. However, in Table 3, All "CADM" should be "CADM-G"; we will correct them in the final version.
> 3. We will revise all the typos, grammar mistakes, and other space redundancy missing throughout the paper, including the reviewer's suggestion
>
> ### Question 5: Clarification on the limitations of the work
>
> In this work, we consider two problems: diversity and adversarial effects in generated images. For the adversarial effects, we characterize the problem as the generated image achieving very high confidence in the conditional class but having minimal features to belong to that class. However, we can observe many generated images that do not have high confidence in the conditional class. For these cases, our proposed method, ProG, can not solve. We hypothesize that the signals from the Diffusion model somehow have dominated the signals from the classifier under some conditions. This negates all the classifier gradients that can help to transform the images into the expected condition.

---

> > ### Author Response · Authors · 2023-08-12
> > **Looking forward to the response from the Reviewer mBBP**
> >
> > Dear Reviewer mBBP,
> >
> > We want to express our sincere gratitude for your valuable feedback on our work. Your insights have greatly contributed to enhancing the quality of our manuscript. In response to your comments, we have taken thorough measures to address each concern:
> >
> > 1. Low improvement on CIFAR-10
> > 2. Details to explain the entropy perspective. Sorry for the equations with index {t, i} in the rebuttals. It should be {i,t} as the reviewer suggested. We will carefully fix this issue in writing.
> > 3. Clarification on Tables 3 and 6.
> > 4. Minor writing issues. We acknowledged several minor issues during the writing and will fix that in the final version. Section 2 will also be rewritten to be more informative, and section 3 will be revised to be slower for readers.
> > 5. Clarification on the limitations of the work: We provided the hypothesis behind the observations and left the problem for future work.
> >
> > Once again, we extend our heartfelt thanks for your invaluable input. Your dedication to the peer-review process has been instrumental in shaping the quality of our manuscript. Please feel free to reach out with any further comments, and we assure you that each concern will be dealt with with the utmost attention.
> >
> > Best regards,
> >
> > Authors #3926

---

> > > ### Comment · Reviewer_mBBP · 2023-08-17
> > >
> > > Thanks for your feedback and I will keep my previous rating.

---

> > > > ### Author Response · Authors · 2023-08-17
> > > >
> > > > Dear Reviewer mBBP,
> > > >
> > > > Thank you for your reply.
> > > >
> > > > Best regards,

---

### Official Review · Reviewer_vxij · 2023-07-05

**Soundness:** 3 good
**Presentation:** 2 fair
**Contribution:** 2 fair
**Rating:** 5
**Confidence:** 4

**Summary:**

In this paper, the authors proposed a new classifier guidance technique for diffusion models named Progressive Guidance (PROG). PROG is an extended version of the classifier guidance method and incorporates the relevant classes' information (beyond the target class alone) to determine the guidance direction, particularly in the presence of noisy images during the early sampling stage. Through extensive experiments on various datasets and diffusion models, the authors demonstrate the effectiveness of PROG in comparison to the standard classifier guidance technique.

**Strengths:**

[1] I think the paper is well written and easy to understand.

**Weaknesses:**

[1] I think the technical novelty of this paper may not meet the standards required for acceptance at NeurIPS.

[2] In my opinion, the proposed PROG technique is only marginally better or comparable to the standard classifier guidance method (according to Tables 1 and 3).

[3] Furthermore, I am not sure about whether the proposed classifier guidance (PROG) is superior to classifier-free guidance in terms of performance and applicability. The image generation community has recently witnessed a rapid shift from class conditional image synthesis to text-to-image synthesis. However, it appears that the proposed classifier guidance method may not be applicable to text-to-image synthesis tasks.

**Questions:**

see above

**Limitations:**

As mentioned in the weaknesses section, I think that the performance improvement of the proposed method over the standard classifier guidance is marginal, and the limited applicability of classifier guidance is a significant limitation of this paper. To strengthen the submission, I recommend that the authors emphasize the strong advantages of their approach compared to standard method (classifier guidance). By highlighting these unique aspects, the authors can make their submission more compelling and is able to address the aforementioned limitations.

---

> ### Author Rebuttal · Authors · 2023-08-09
>
> ### Question 1: The novelty of the work
> Since the reviewer does not mention the reason why our work is not novel, we summarize our three main novelties below:
>
> [**Nov1**] : Detects and justifies **diversity suppression** problem due to classification gradient. As far as we know, this problem has **NOT been investigated** before. In GANs, the mode collapse problem that causes the lack of diversity has a different essence than the diversity suppression problem. The problem can be observed in Figure 2 (main paper) and Figures 5, 6, and 7 (Appendix). => **novel**
>
> [**Nov2**]: Quantifies the problem of **adversarial effect** due to classification gradient as observed in Figure 3 (main paper), Figure 8, 11, 12, 13, 14 (Appendix). As far as we know,  this problem has **NOT been investigated** before. Although in [1], the authors have mentioned the adversarial effect of the classifier guidance, this problem has never been justified and solved. => **novel**
>
> [**Nov3**]: Develop an intuitive approach addressing both [**Nov1**] and [**Nov2**] simultaneously. First, mitigate adversarial effects by utilizing gradients from diverse classes, minimizing noise associated with one class. This is exemplified in Figure 3. Second, incorporate information from other classes to prevent diversity suppression and overemphasis on a single class, as illustrated in Figure 2. Lastly, introduce progressive guidance to enhance primary conditions toward sampling completion. Due to its unique philosophy and absence in prior literature, our proposal stands as a **novel** technical contribution.
>
> ### Question 2: The significance of the proposed method compared to vanilla classifier guidance.
>
> In the main paper and Appendix, we offer **5 significant improvements** compared to vanilla guidance:
>
> [**SIG1**] *Diversity quantitative* improvement: in Figure 4 (up to **45% improvement FID**, and **35% on Recall**); right Figure in Table 4 (improve up to **22% on FID** and **22.5% on Recall**), Figure 9 (improve up to **28.5% on FID** and **40% on Recall**) , Figure 10 (up to **50% on FID** and **37.5% on Recall**) in Appendix . Since the gap is large, it is **significant** enough to observe.
>
> [**SIG2**] *Quantitative robust features* improvement as detailed in Table 2, 4. The improvement is clear (around **6% in Table 2** and **3% in Table 4**). Thus,  this improvement is considered **significant**.
>
> [**SIG3**] *Diversity qualitative* improvement as detailed in Figure 2, 5 (right), 6 (right), 7 (right). The improvement can be observed by human eyes. As a result, it should be **significant**.
>
> [**SIG4**] *Qualitative robust features* improvement as detailed in Figure 3(right), Figure 8, Figure 11, 12, 13, 14. The improvement can be observed by human eyes, so we believe it is **significant**.
>
> [**SIG5**] *Quantitative improvement over some traditional generative metrics* such as FID, sFID and IS. The improvement is shown by percentage as in Table R8 and R9.
>
> | | FID imp.  | sFID  imp. | IS  imp. |
> |------|--|----|--
> | ImageNet 64x64   | 19.37%/ 20% / 7% | 30.5% /  4% / 6.27% | 0% / 12.54% / 0% |
> | ImageNet 128x128 |  7% | 0.1% | 11% |
> | ImageNet 256x256 | 1% / 6%   | 2.5% /18.5% | 2.5% / 18.5%|
> | CIFAR-10| 1.4%| 0.4% | -1.35%|
>
> Table R8: (Table 1 in main paper) Except for t the CIFAR-10,  we achieved significant improvement.
>
> |  | FID  imp.| sFID   imp.| IS  imp.|
> |----|--|--|--|
> | ImageNet 64x64   | 2.5%| 4.30% | 6.42%	     |
> | ImageNet 128x128 | 4.3% | 0% | 9.40%        |
> | ImageNet 256x256 | 3% / 0.88%| 0%| 8.77% / 1.46%|
>
> Table R9:  (Table 3 in main paper) The improvement is clear on FID and IS on all databases and sFID on ImageNet64x64.
>
> In Table R8, Table R9, [**SIG1**], [**SIG2**], [**SIG3**], [**SIG4**], significant improvements are evident, except for CIFAR-10. The limited progress on CIFAR-10 is due to its classes having little shared information, reducing ProG's impact.
> ### Question 3.1: Extend ProG to Text-to-Image problem:
>
> We have successfully extended our proposed ProG to improve guidance for Text-to-Image. The experimental settings and results are in the joint rebuttal to all reviewers.
>
> ### Question 3.2: Text-to-image (Text2Img) condition vs. class condition (ClsCon)
> The reviewer mentions a shift from ClsCon to Text2Img, but recent works in both years (2022 and 2023) indicate continued interest in ClsCon [2-9]. Both ClsCon and Text2Img are crucial in generative models, and it's unjust to prioritize one over the other.
>
> ### Question 3.3: Classifier guidance vs. Classifier-free guidance
> The reviewer mentions the limited application of classifier guidance compared to classifier-free guidance. However, classifier guidance offers more flexibility:
>
> * Training: Classifier-free needs complete retraining of diffusion models for new conditions, while classifier guidance only requires classifier updates.
> * Sampling: Classifier guidance works with unconditional or conditional diffusion, unlike classifier-free, which requires both.
> * Computational cost: Classifier-free is computationally expensive (Table R6).
> * Extendability: Both  can extend to various conditions, e.g., Text to image.
>
> | Model| Sampling cost (GPU hours) |
> |:--:|:----:|
> | Diffusion  |  236 |
> | Vanilla guidance |  341  |
> | ProG guidance | 341|
> | Classifier-free guidance | 487|
>
> Table R6: Computational cost to generate 50000 images with 256x256 resolution.
>
> We summarize the features of each guidance method in Table R7.
>
> | | **Training flex.** | **Sampling flex.** | **Low cost** | **Robustness** | **Diversity** | **Extendibility** |
> |--|:-----:|:--:|:--:|:---:|:--:|:----:|
> | Vanilla guidance   | yes | yes | yes | no | no | yes  |
> | Classifier-free guidance | no  | no   | no  | yes  | no | yes|
> | ProG   | yes  | yes | yes  | yes | yes | yes|
>
> Table R7: As we can see, the main reason for the popularity of classifier-free guidance is its robust features. However, ProG can combine all the advantages of Vanilla and Classifier-free guidance in one unified scheme.

---

> > ### Author Response · Authors · 2023-08-12
> > **Looking forward to the response from the Reviewer vxij**
> >
> > Dear Reviewer vxij,
> >
> > Thanks for your valuable time in commenting on our work. We have analyzed your comments thoroughly and carefully and answered in detail with experimental backup/evidence for what we have claimed:
> > 1. The novelties of our work. Since the observations on **diversity suppression** and **adversarial effects** are not available in any previous research, we consider them novel. Furthermore, the **proposed ProG** is carefully designed to solve the problem based on the observations, which never appeared in any previous work before, and should also be a novel technique.
> > 2. We provided evidence that our proposed ProG is significantly better than vanilla classifier guidance in 5 ways. We hope the evidence we provide solves your concern about the significance of our proposed method.
> > 3. We extended our ProG to improve Text-to-Image guidance which is the last main concern of the Reviewer. We hope that from the improvement in results, the Reviewer can reconsider the contributions of our works.
> >
> > We hope that from the evidence/experimental results we provide, the Reviewer can reconsider our work's novelty, significance, and application for the research community. If the Reviewer still has any concerns, please do not hesitate to let us know. We are happy to address them.
> >
> > Best regards,
> >
> > Author #3926

---

> ### Author Response · Authors · 2023-08-17
> **Looking forward to the response from Reviewer vxij**
>
> Dear Reviewer vxij,
>
> We have tried our best to address all the concerns and provided as much evidence as possible. May we know if our rebuttals answer all your questions?
>
> Best regards,
>
> Author #3926

---

> > ### Comment · Reviewer_vxij · 2023-08-18
> >
> > I appreciate your thorough response. The general feedback provided and the subsequent response from the authors have satisfactorily addressed some of my concerns, particularly regarding the novelty aspects. Nevertheless, I still believe there are certain points that require further clarification from the authors.
> >
> >
> > (1) **Extending ProG to Text-to-Image Problem:**
> >
> > In the general feedback, the authors discuss the potential extension of ProG for guiding text-to-image synthesis. However, I've expressed my concerns into three points. Firstly, the FID values presented in Tables R1 and R2 do not align with those in the GLIDE paper (please see Figure 6 and Table 2 of the GLIDE paper). Notably, the FID score of ProG trails behind that of Classifier-Free Guidance (CFG; 256px FID = 12.89). Moreover, unlike CFG, the implementation of ProG necessitates pre-trained noise-aware classifiers, a constraint that restricts its applicability. This limitation is likely why the authors evaluated their method using GLIDE rather than Stable Diffusion. Lastly, as highlighted by Kynkäänniemi et al. [R1], using a classifier can introduce information leakage to FID, thereby leading to improper comparisons. Consequently, I believe that the authors should conduct a user study to thoroughly validate whether ProG genuinely enhances image quality and diversity more effectively than CFG.
> >
> > (2) **Additionally, I cautiously suggest that the authors should consider refining the arguments below:**
> >
> > **Training: Classifier-free needs complete retraining of diffusion models for new conditions, while classifier guidance only requires classifier updates.**
> >
> > => Conversely, the classifier-guided approach necessitates training an expensive noise-aware classifier. Particularly for text-to-image tasks, training such a classifier could potentially complicate the entire diffusion pipeline.
> >
> > **Sampling: Classifier guidance works with unconditional or conditional diffusion, unlike classifier-free, which requires both.**
> >
> > => Could the model trained using classifier-free guidance be employed for both unconditional and conditional image synthesis?
> >
> > **Computational cost: Classifier-free is computationally expensive (Table R6).**
> >
> > => I suspect the authors have overlooked the computational expenses incurred by ChatGPT. If the authors were to factor in the ChatGPT-related costs, would the proposed approach still maintain its computational efficiency in comparison to other methods?
> >
> > **Extendability: Both can extend to various conditions, e.g., Text to image.**
> >
> > => I'm uncertain whether ProG is applicable to text-to-image synthesis.
> >
> > Due to these considerations, I have modestly increased my score by one point. However, I am not yet prepared to advocate for the acceptance of this paper.
> >
> > [R1] Kynkäänniemi, T., Karras, T., Aittala, M., Aila, T., & Lehtinen, J. (2022). The Role of ImageNet Classes in Fr\'echet Inception Distance. arXiv preprint arXiv:2203.06026.

---

> > > ### Author Response · Authors · 2023-08-18
> > >
> > > Dear Reviewer vxij,
> > >
> > > Thank you for your reply. We address your two main concerns below:
> > >
> > > ### Concern 1: Extending ProG to Text-to-Image Problems
> > > > 1. "The Zero-shot FID values presented in Tables R1 and R2 do not align with those in the GLIDE paper."
> > > >> **Answer**: The main reason is that the authors of the GLIDE paper *do not release the full pretrained model* due to their **privacy concerns**. As a result, they only release a reduced version of the pretrained model for verification; hence, the quality is limited. This is stated in the GLIDE paper's abstract and section E (Appendix).
> > > > 2. "unlike CFG, the implementation of ProG necessitates pre-trained noise-aware classifiers, a constraint that restricts its applicability. This limitation is likely why the authors evaluated their method using GLIDE rather than Stable Diffusion."
> > > >>**Answer:**: We respectfully disagree with this point of view. After the noise-aware classifier is trained, it can be applied to any diffusion model with the same latent space size. The main reason that noise-aware CLIP can not be applied to Stable Diffusion is that its latent size is larger than Stable Diffusion's latent size. This does not mean noise-aware classifier/CLIP can not work with other diffusion models. In reply to the Reviewer cJ7t, we show a case where the noise-aware CLIP from the GLIDE paper [11] can be applied to the ADM diffusion model in [10] without any difficulties and achieves some improvement. On the other hand, given a diffusion model which has been trained to do classifier-free guidance, it has to be stuck with that diffusion model and can not work with any other diffusion models. For example, Stable Diffusion can not work with DiT [5] even when they share the exact latent diffusion mechanism.
> > > > 3. "as highlighted by Kynkäänniemi et al. [R1], using a classifier can introduce information leakage to FID, thereby leading to improper comparisons".
> > > >> **Answer**: The diffusion model and noise-aware CLIP (the equivalent model for classifier) in GLIDE do not use any classification or label information, especially classification information from ImageNet. Eq.r1 shows the sampling equation where we utilize the matching between image embedding and text embedding without any classification gradient through input. As a result, the result is not affected by the information leakage to FID.
> > >
> > > ### Concern 2: Clarify the argument:
> > >
> > > > 1. "The classifier-guided approach necessitates training an expensive noise-aware classifier. Particularly for text-to-image tasks, training such a classifier could potentially complicate the entire diffusion pipeline."
> > > >> **Answer**:
> > > >>>* The training of a noise-aware classifier is similar to training a classifier. The only difference is the augmentation step which we add the Gaussian noise to the image before feeding it to the classifier. As a result, this training process will be much cheaper than training a diffusion model. This is because training a discriminative task is always cheaper than training a generative task.  On the other hand, training a model for classifier-free guidance must go with the training generative model, which is extremely expensive and unnecessary when a new condition is introduced. e.g., more text, more classes,...
> > > >>>* Furthermore, the training of the noise-aware classifier is separate from the training of the diffusion model. As a result, it can not complicate the entire diffusion pipeline.
> > > > 2. "Could the model trained using classifier-free guidance be employed for both unconditional and conditional image synthesis?"
> > > >> **Answer**: The classifier-free guidance can only be applied when the diffusion model is trained to incorporate the conditional information and the null condition. But the classifier guidance imposes no restrictions on the diffusion model, and it can be applied to any diffusion model, either trained to be conditional or unconditional or even a combination between condition and null condition. The scenario that classifier-free can not be applied is not rare in practice. What if, in case, we don't have conditions before training the diffusion model? It is impossible to train to combine conditional information and null condition with a diffusion model for classifier-free guidance later.
> > > > 3. "I suspect the authors have overlooked the computational expenses incurred by ChatGPT."
> > > >> **Answer**: ChatGPT-related cost is the pre-processing cost, where we only do it once for one set of labels. It is similar to the cost of **data collection**. We don't count data collection time as the cost for sampling/training. On the other hand, the expensive cost of classifier-free guidance comes from forwarding through the diffusion model twice every each timestep.
> > > > 4. Extendability:
> > > >> **Answer**: We have answered already in the Concern 1 and Table R1 and R2 can show that ProG can extend for Text-to-Image guidance.
> > >
> > > May we know if we have solved your concern?
> > >
> > > Best regards,

---

> > > > ### Comment · Reviewer_vxij · 2023-08-19
> > > >
> > > > Thank you for reviewer's comment. You answer has resolved all my concerns. I am willing to increase my score 4 to 5.

---

> > > > > ### Author Response · Authors · 2023-08-20
> > > > >
> > > > > Dear Reviewer vxij,
> > > > >
> > > > > Thank you for your comments and advocacy for accepting our work.
> > > > >
> > > > > Best regards,
> > > > >
> > > > > Author #3926

---

### Official Review · Reviewer_xoqA · 2023-07-07

**Soundness:** 3 good
**Presentation:** 3 good
**Contribution:** 3 good
**Rating:** 6
**Confidence:** 4

**Summary:**

To tackle the generative issue of low diversity and artifacts of classier guidance for conditional diffusion generation, this paper proposes an entropy view for calculating the conditional score gradient. The proposed method proposes two modifications for classifier guidance, 1) exchanging one-hot class labels with soft-labels based on class similarity. 2) progressive score weights for different time steps. And the proposed achieves better results than the baseline method and classifier-free guidance method.

**Strengths:**

1. The proposed method is simple and straightforward yet effective to achieve better generative results.
2. The proposed method suggests an entropy view to review the classifier guidance method is interesting. In the real dataset, we should consider the real conditional distribution of the label given image, which is neglected by the vanilla one.

**Weaknesses:**

1. Although the proposed method is simple and straightforward, the flexibility would be restricted for different modalities of conditions such as (text and segmentation map), which can be easily addressed via classifier-free guidance. Is there any solution for classifier guidance with flexible conditional labels (different modalities)?

**Questions:**

Please refer to the weakness.

**Limitations:**

This paper does not discuss the limitations. However, the proposed method might not be general for conditional generation with diverse modalities of conditional labels (etc. text, segmentation map).

---

> ### Author Rebuttal · Authors · 2023-08-09
>
>
>
> ### Question 1: Extend the proposed method into Text-to-image guidance.
> GLIDE[11] is an easy extension of classifier guidance for text-to-image guidance. The sampling equation for GLIDE is shown below:
>
> $$x_{t-1} = \mu _t + \sigma _t * \mathbf{z} + s \sigma_t^2 \nabla _{x_t} (f(x_t) . g(c))\quad \quad _{(r1)}$$
>
> $f(x_t)$ is the image embedding vector and $g(c)$ is the text or description embedding vector. Equation (r1) is mostly similar to equation (3) in our main paper; the only difference is the gradient term resulted from the similarity between two embedding vectors instead of the classification gradient as in the main paper.
>
> We add new experiments to apply ProG for GLIDE in equation (r1) following two scenarios:
> 1. Given one caption, we will utilize a random set of 1000, 5000, or 10000 captions to act as relevant information during sampling. we have: $$g(c) = \sum_{i=0}^{N+1} s_i g(c_i)\quad\quad_{(r2)}  $$ with $i = 0$ is the index of the primary caption, and $i \neq 0$ represents other captions. The initial  values of $s_i$ are set as: $$s_i = \frac{g(c_0) . g(c_i)}{\sum^{N+1}_{j=0} g(c_0) .g(c_j)} \quad \quad _{(r3)}$$ The value of $s_i$ is progressively updated during sampling as in section 3.2 in the main paper. This scheme is named **GLIDE-ProG**
> 2. Given one caption, we use four other captions that have the same meaning as the original caption but different words. Since four other captions, all have the same meaning; we have different strategies to set the $s_i$ values: $$s_i = \begin{cases}a, \quad \text{if } i = 0\\\ \frac{1-a}{4}, \quad \text{otherwise} \end{cases}$$, with $0\leq a \leq 1$ is hyperparameter, we try with $a=0.3$. This method is named **GLIDE-ProGsim**
>
> We set up an evaluation like GLIDE [11] to evaluate zero-shot FID on MS-CoCo. Note: 4 additional equivalent captions of GLIDE-ProGsim are taken from the set of captions available for each image in MS-Coco captions. 1k, 5k, and 10k captions are randomly sampled from the MSCoco training set. Table R1 and Table R2 show the evaluation results:
>
> | | zero-shot FID | computational cost (GPU hours)|
> |:----:|:------:|--------|
> | GLIDE |    24.80  | 34.27  |
> | GLIDE-ProG w N=1k |  23.47| 34.66 |
> | GLIDE-ProG w N=5k |23.50|	34.83 |
> | GLIDE-ProG w N=10k|	 **23.31**|34.83|
> | GLIDE-ProGsim | 23.87 |34.84 |
>
> Table R1: MSCoCo64x64 zero-shot FID evaluation where 30000 captions are sampled from the MSCoco validation set.
>
>
>
>
> |               	| zero-shot FID | computational cost (GPU hours) |
> |:----------------:|:-------------:|--------------------|
> | GLIDE         	|      34.80    |      38.45              |
> | GLIDE-ProG w N=1k |      32.55    |    45.50                |
> | GLIDE-ProG w N=5k |	   32.37	|				45.80	 |
> | GLIDE-ProG w N=10k|	   32.28	|				46.10	 |
> | GLIDE-ProGsim 	|      **31.91**    |   46.23                 |
>
> Table R2: MSCoCo256x256 zero-shot FID evaluation where 30000 captions are sampled from the MSCoco validation set.
>
> **Conclusion**: From Table R1 and Table R2, the ProG scheme helps significantly improve the performance of text-to-image guidance in different scenarios with low additional computational costs. Given the many captions available, we can use the first scenario to improve the generated images. Otherwise, the second scenario is also very easy to implement. The additional captions can be gathered from Large Language Models (LLMs) to generate images.
>
> ### Question 2: The flexibility of classifier guidance.
>
> Like classifier-free guidance, classifier guidance can extend to different modalities. More than that, classifier guidance also has many advantages:
>
> * Training: Classifier-free needs complete retraining of diffusion models for new conditions, while classifier guidance only requires classifier updates.
> * Sampling: Classifier guidance works with unconditional or conditional diffusion, unlike classifier-free, that requires both.
> * Computational cost: Classifier-free is computationally expensive (Table R6).
> * Extendability: Both support or can extend to various conditions, e.g., Text to image.
>
> | Model| Sampling cost (GPU hours) |
> |:--:|:----:|
> | Diffusion  |  236 |
> | Vanilla guidance |  341  |
> | ProG guidance | 341|
> | Classifier-free guidance | 487|
>
> Table R6: Computational cost to generate 50000 images with 256x256 resolution.
>
> We summarize the features of each guidance method in Table R7.
>
> | | **Training flex.** | **Sampling flex.** | **Low cost** | **Robustness** | **Diversity** | **Extendibility** |
> |--|:-----:|:--:|:--:|:---:|:--:|:----:|
> | Vanilla guidance   | yes | yes | yes | no | no | yes  |
> | Classifier-free guidance | no  | no   | no  | yes  | no | yes|
> | ProG   | yes  | yes | yes  | yes | yes | yes|
>
> Table R7: As we can see, the main reason for the popularity of classifier-free guidance is its robust features. However, ProG can combine all the advantages of Vanilla and Classifier-free guidance in one unified scheme.

---

> > ### Comment · Reviewer_xoqA · 2023-08-18
> >
> > Hi,
> >
> > Thanks for the reply, my question is addressed.

---

> > > ### Author Response · Authors · 2023-08-18
> > >
> > > Dear Reviewer xoqA,
> > >
> > > Thank you very much for your comments and strong support for us.
> > >
> > > Best regards,
> > >
> > > Author #3926

---

> ### Author Response · Authors · 2023-08-12
> **Looking forward to the response from the Reviewer xoqA**
>
> Dear Reviewer xoqA,
>
> Thanks for your valuable and thoughtful comments about our work. In the rebuttal, we solved the only problem that the Reviewer was concerned about using classifier guidance and our proposed ProG for different conditional labels. We have shown that classifier guidance/ProG can easily be extended to Tex-to-Image guidance by replacing the classifier with a CLIP model. Furthermore, we generalized the author's concern by showing the flexibility of classifier guidance compared to classifier-free guidance. By this, we hope that the concern about the flexibility of our method is solved thoroughly.
>
> We hope that our rebuttals can gain strong support from the Reviewer. If the Reviewer still has any other concerns, we look forward to answering them.
>
> Best regards,
>
> Author #3926

---

> ### Author Response · Authors · 2023-08-17
> **Looking forward to the response from the Reviewer xoqA**
>
> Dear Reviewer xoqA,
>
> Thank you for your valuable comments on our work. We sincerely hope that our rebuttals have addressed all your concerns. Kindly let us know if you have any other concerns, and we will do our best to address them.
>
>
> Best regards,

---

> ### Comment · Area_Chair_gqak · 2023-08-17
> **Please take a look at authors' responses, Thanks!**
>
> Dear Reviewer, please take a look at authors' responses and other reviewers' comments, Thanks!

---

### Official Review · Reviewer_cJ7t · 2023-07-07

**Soundness:** 3 good
**Presentation:** 3 good
**Contribution:** 3 good
**Rating:** 5
**Confidence:** 4

**Summary:**

This paper proposes to inject the graident of other clasess to improve the diversity for conditional sampling of diffusion model.

**Strengths:**

1. The general idea is simple, the gradient of the classifier tends to use the most discriminative feature and thus hurts the performance, so we use the gradient of other classes in early phases to improve the diversity.

2. The paper also provides some entropy arguments, which partly justifies the method.

3.Extensive experiments are conducted and the performance gain is clear.

**Weaknesses:**

1. Missing details. How many steps are used for the eq 4?. "In the later sampling stage, we progressively enhance gradients to refine the details 13 in the image toward the primary condition" Do you just use classifier guidance after some steps?

2. Authors use CLIP embedding to compute the similarity between different classes. Can we use clip gradient to guide the generation directly (insert the graident of CLIP similarity between the generated image and target class, say "dog" )  in DDIM steps? Will that also improve the diversity as CLIP has seen many different dog images. Some comparions are needed.


**Questions:**

as above

**Limitations:**

as above

---

> ### Author Rebuttal · Authors · 2023-08-09
>
>
>
> ### Question 1: Elaborating on details
>
> 1. For equation 4, the number of timesteps is 1000. However, similar to [10], we adapt the respace to 250 timesteps (skip four iterations each time). All other hyperparameters are detailed in Table 10 (Appendix)
> 2. "*Do you just use classifier guidance after some steps*"? -> No, we use classifier guidance from the start to the end.
> 3. "*In the later sampling stage, we progressively enhance gradients to refine the details in the image toward the primary condition*." We mean that when we start, the gradients from other classes are noisy. Following the iterations, the gradients from different categories are reduced, and the gradient from the primary class is emphasized (more weight).
>
> ### Question 2: Use CLIP embedding for guidance
>
> This would be an exciting idea since it allows using off-the-shelf information to improve generated images from diffusion models. We set up the experiments as below:
> 1. Use pretrained CLIP from GLIDE [11] (an extension of classifier guidance for text-to-image guidance)
> 2. Use the pretrained Diffusion model (ADM) from [10] as a model to generate ImageNet
>
> The method is named CLIP guidance. We evaluate on ImageNet64x64. The results are shown in Table R5:
>
>
> |                  | FID | sFID | Recall |
> |:----------------:|:---:|------|--------|
> | Vanilla guidance | 6.40  | 9.67    | 0.54   	     |
> | ProG guidance    | **5.16**  | **6.72**    | 0.56   	 |
> | CLIP guidance    | 8.18  | 9.4     | **0.59**   	 |
> | Diffusion w/o guidance| 9.95| 6.58| 0.65|
>
> Table R5: ImageNet64x64. Comparison between Vanilla guidance, ProG guidance, and CLIP guidance.
>
> The results show that the ProG achieves the best FID/sFID among the guidance schemes. However, the Recall value of CLIP guidance is the highest one. This indicates the ProG provides the diversity, but in the original data distribution, while the CLIP guidance could help to achieve better variety outside of the data distribution. It must be noted that the FID obtained from the CLIP guidance is better than the original diffusion model without guidance. This is concrete evidence that the guidance from off-the-shelf information is working. However, it might need a lot of experiments to observe the fundamental problems inside. We like the idea and would be interested in further developing this idea in future work.

---

> > ### Author Response · Authors · 2023-08-12
> > **Looking forward to the response from Reviewer cJ7t**
> >
> > Dear Reviewer cJ7t,
> >
> > Thank you very much for your insightful comments and interesting suggested ideas.
> >
> > We have addressed your primary concerns about the missing description details in our manuscripts. Besides, we are also very grateful to receive an exciting idea from the reviewer. We obtained some preliminary results which can help open a new door to solve the problem of diversity in the future.
> >
> > If you have further comments regarding the work, we are happy to address yours. We hope to gain your strong support for our work.
> >
> > Best regards,
> >
> > Authors #3926

---

> ### Author Response · Authors · 2023-08-17
> **Looking forward to the response from Reviewer cJ7t**
>
> Dear Reviewer cJ7t,
>
> Thank you for your valuable comments on our work. We sincerely hope that our rebuttals have addressed all your concerns. Kindly let us know if you have any other concerns, and we will do our best to address them.
>
> Best regards,

---

> > ### Comment · Reviewer_cJ7t · 2023-08-17
> >
> > Thanks for your response. I read reviews from other reviewers and the responses. I share the same novelty concern with other reviewers. But the method is simple and works well on different datasets. So, I will keep my score.

---

> > > ### Author Response · Authors · 2023-08-18
> > > **Thank you for your reply, we actually addressed all the concerns related to novelty from others.**
> > >
> > > Dear Reviewer cJ7t,
> > >
> > > Thanks for acknowledging the strength of our paper which are effectual/fruitful, simple and easy to implement. This is also well-recognized by Reviewers zai6, zHLG, xoqA, and mBBP.
> > >
> > > Regarding the novelty concerns, there are only two reviewers raise novelty concerns, and we have well addressed the concerns:
> > > >* For **Reviewer vxij**:
> > > >> * We don't know the reasons behind Reviewer vxij's novelty concerns since no reasons are provided in the comment. However, we address this comprehensively in our vxij rebuttal's Question 1, highlighting evidence for three key novelties: **diversity suppression**, **adversarial effects**, and **ProG method**.
> > > >>*  The **diversity suppression** and **adversarial effect novelties** are recognized by **Reviewer mBBP** through the comment:
> > > >>> "This paper has clear and well-established motivations, working on two important problems in classifier guidance encountered by the community, the adversarial effect and diversity suppression. This can be a good contribution to the community.",
> > > >> * **Reviewer zHLG** recognizes the **novelty of the proposed method** through comments
> > > >>>" This work proposes a novel way to improve the classifier guidance method."
> > > >> * You and other reviewers all agree that the strength of the proposed method is simple and effective.
> > > >* For **Reviewer zai6**: The main novelty concern is the lack of novel insights in the main paper. We realized that most of the concerns could be solved by the Appendix file (We could not put everything in the main paper due to page limits). We have addressed this concern thoroughly:
> > > >> * First, In our Appendix, we have provided at least **7 insights**, which are matched with the Reviewer's suggestion to do observations on the **subset of the dataset** instead of dataset level. The details are in the answer to Question 3 of Reviewer zai6's Rebuttal.
> > > >> * Second, we investigated the sensitivity of $\\gamma$ with the FID score in Table 5 of our submission. This is also matched with the Reviewer's suggestion about the **sensitivity of the method**.
> > > >> * Lastly, we include the correlation table between $\\gamma$ and the accuracy of the classifier performance in Table R3 in the Question 1 to answer the concern about **sensitivity of the method regarding the classifier performance**
> > >
> > > May we know if we have solved your novelty concern? If not, can you help point out why you think our work is not novel? Your insights would not only assist us in addressing your concerns more effectively but would also contribute to the enhancement of our future work.
> > >
> > > Best regards,
> > >
> > > Author #3926

---

> > > > ### Author Response · Authors · 2023-08-18
> > > >
> > > > Dear Reviewer cJ7t,
> > > >
> > > > **Reviewer vxij** and **Reviewer zai6** have confirmed that we have already solved the novelty concern.
> > > > > * The Reviewer vxij confirmed it through the comment: "The general feedback provided and the subsequent response from the authors have satisfactorily addressed some of my concerns, particularly regarding the novelty aspects.",
> > > > > * and the Reviewer  zai6 confirmed it by commenting: "By updating the paper with different empirical design choices w.r.t sensitivity of classifier, $\\gamma$, and other additional experiments, it will be valuable to the community."
> > > >
> > > > May we know if you still have concerns about the novelty of our work? If your concern persists, please let us know so that we can address your concern.
> > > >
> > > > Best regards,
> > > >
> > > > Author #3926

---

### Official Review · Reviewer_zHLG · 2023-07-10

**Soundness:** 3 good
**Presentation:** 4 excellent
**Contribution:** 3 good
**Rating:** 7
**Confidence:** 3

**Summary:**

This work points out that diffusion models with classifier guidance only focus on the given category but ignore the other relevant category information. Thus, this work proposes the Progressive Guidance (PG) method to address two problems, i.e., lack of diversity and the adversarial effect (samples have high scores but poor visual quality). The proposed method uses progressive gradients from the class dimension and the diffusion temporal dimension to change the gradient of the classifier guidance on a single condition. In terms of the class dimension, PG allows gradient-assisted conditions to be generated for other class information related to a given class. In terms of diffusion temporal dimension, the weight of the gradient also changes over time. The experimental results show that PG improves image quality, sample diversity, and robustness compared with the competing methods.

**Strengths:**

1. The paper is well-organized and easy to follow.

2. This work proposes a novel way to improve the classifier guidance method.

3. The proposed dramatic diversity and non-robustness feature construction approaches show desirable robustness than the commonly used baseline method.

4. The proposed method can be combined with powerful backbone networks to achieve favorable performance.

**Weaknesses:**

1. Though the theoretical analyses are convincing, the experimental results show that the proposed method sometimes underperforms the classifier-free guidance. The authors should explain this point to verify the effectiveness of their proposed method.

2. In Section 3, the analyses could be more convincing if more evidence is provided. In addition, the presentation of Sections 3.1 and 3.2 should be improved to make it clear.

3. The computational costs should be clarified, considering that the diffusion models are usually expensive to implement for producing high-resolution images.


**Questions:**

Please see my comments above.

**Limitations:**

Please see my comments above.

---

> ### Author Rebuttal · Authors · 2023-08-09
>
> ### Question 1: Clarify the performance of classifier guidance in some cases.
>
> Sometimes, the proposed classifier guidance performs poorly than classifier-free guidance, such as FID/sFID and Robustness score on ImageNet256x256. However, the two methods can be considered comparable on this case due to several reasons:
> * The gap between FID/sFID between the two methods is not significant (3.84 vs. 3.76) (~0.08)
> * The gap between Robustness score between the two methods is not so significant ( 86.60 vs 87.14) (~0.54%)
> * ProG has a much **better level of diversity** than classifier-free guidance, as shown in Figure  16(right) in the Appendix. FID and Recall values are much higher than classifier-free guidance when $w$ is large.
> * ProG has a much **lower computational cost** compared to classifier-free guidance, as shown in Table 8 (Appendix). Classifier guidance only needs 341 GPU hours to generate 50000 images compared to 487 for classifier-free guidance.
>
> Besides the comparability in the performance between the two methods, classifier guidance has several advantages over classifier-free guidance in terms of application:
> 1. **Training flexibility**:  Classifier-free guidance used the information from the conditional diffusion model. As a result, when the condition is modified, or a new condition is available, there would be no light-cost solution but to train the whole expensive diffusion model. On the other hand, since classifier guidance allows separate classifiers. The update in condition allows updating on the classifier only without re-training expensive diffusion models.
> 2. **Sampling flexibility**: We can apply the classifier-free technique only when we have both unconditional and conditional diffusion models simultaneously (can be separated or joined). However, classifier guidance can be used given solely unconditional diffusion, solely conditional diffusion, or both.
>
> ### Question 2: Analyse in section 3 should be improved, and more evidence should be provided.
>
> Due to the page limit, most of the evidence from section 3 is put into the Appendix. We will return this evidence to the main paper for a better reading experience. In detail, we have:
>
> For **diversity suppression**, we have done on many types of breeds in ImageNet:
>
> + [**EVD1**] Problem of front-face features collapsing of vanilla classifier guidance in Figure 5 (for Brittany Spaniel), Figure 6 (for English Springer), and Figure 7(for Welsh Springer Spaniel).
> + [**EVD2**] Problem of collapsing all images into single pose features in Figure 5 (for Brittany Spaniel)
> + [**EVD3**] Problem of green grass background in vanilla guidance for some types of dog Figure 7(for Welsh Sprinter Spaniel)
>
> For **non-robust features construction**, we did mainly on some types of breeds and leopard class of ImageNet  :
>
> + [**EVD4**] Problem of losing background in vanilla guidance in Figure 8.
> + [**EVD5**] Problem of non-robust features construction of vanilla guidance in Figure 11 (for ImageNet64x64) and Figure 12, 13, 14 (For ImageNet256x256)
>
> ### Question 3: The computational cost
>
> We have mentioned the computational cost in Table 8 (Appendix). To clarify the computational cost, we will revise Table 8 into the below table, where Diffusion computational cost and vanilla guidance cost will be detailed.
>
> |             Model        | Computational cost (GPU hours) |
> |:------------------------:|:------------------:|
> | Diffusion                |         236        |
> | Vanilla guidance    |         341        |
> | ProG guidance    |         341        |
> | Classifier-free guidance |         487        |
>
> Table R4: Computational cost to generate 50000 images with 256x256 resolution.

---

> > ### Author Response · Authors · 2023-08-12
> > **Looking forward to the response from the Reviewer zHLG**
> >
> > Dear Reviewer zHLG,
> >
> > Thank you for your thoughtful and insightful comments on our work. We believe that these comments help us to strengthen our submission.
> >
> > In the rebuttal, we have addressed all of your three concerns:
> > 1. Clarify the performance of our method over classifier-free guidance
> > 2. Provide more information for section 3. Section 3.1 and section 3.2 will be revised in our final manuscript.
> > 3. We provided the computational cost comparison between guidance methods.
> >
> > If you still have further requests or concerns, please do not hesitate to let us know. We will deal with the concerns with the utmost attention.
> >
> > Best regards,
> >
> > Author #3926

---

> > > ### Author Response · Authors · 2023-08-17
> > > **Looking forward to the response from Reviewer zHLG**
> > >
> > > Dear Reviewer zHLG,
> > >
> > > Thank you for your valuable comments on our work. We sincerely hope that our rebuttals have addressed all your concerns. Kindly let us know if you have any other concerns, and we will do our best to address them.
> > >
> > > Best regards,

---

> > > > ### Comment · Reviewer_zHLG · 2023-08-18
> > > >
> > > > Thanks for your responses. Now my concerns have been addressed, and I believe this paper is a decent work.

---

> > > > > ### Author Response · Authors · 2023-08-18
> > > > >
> > > > > Dear Reviewer zHLG,
> > > > >
> > > > > Thank you very much for your comments and strong support for our work.
> > > > >
> > > > > Best regards,
> > > > >
> > > > > Author #3926

---

### Official Review · Reviewer_zai6 · 2023-07-11

**Soundness:** 3 good
**Presentation:** 2 fair
**Contribution:** 2 fair
**Rating:** 7
**Confidence:** 4

**Summary:**

In this work authors propose to address challenges of classifier guidance Diffusion models and propose progressive-guidance where during sampling/reverse diffusion process. Initial iterations of reverse process receives classifier gradient from multiple relevant classes instead of just target class so that more relevant features can be retained. Authors also illustrate this enables generated images to be more diverse as features in intial iterations need not 'ONLY' be purely discriminative for current target class of interest.

**Strengths:**

Authors propose a simple and useful method to improve sample diversity of Diffusion models within classifier guidance setting and in appendix they show sample diversity is on-par with classifier-free guidance.

Progressive Guidance makes sense more generally from generation perspective too, as within generative paradigm we first sample higher level semantics which is not very fine-grained then conditioned on that we sample latent variables/features relevant to fine grained details so it does make sense that we don't want to hyper-focus on one-particular class but it probably depends on complexity of class-taxonomy of particular dataset.

**Weaknesses:**

Core idea in paper is easy to follow but lack novel insights or significant contributions.

Few suggestions in terms of details and writing:

Though this paper follows classifier guidance from previous literature, it might be useful to summarize noise-aware classifier performance at difference noise-levels, especially from what iteration is classifier gradient incorporated in sampling process.

What is value of gamma at different sampling steps w.r.t number of inference steps and schedulers to easily interpret results and setting.

What is actual guidance scale value across sampling steps/reverse diffusion, more specifically at \gamma = 0.04 at what stage of sampling is classifier guidance focusing on one-hot vector. If you can explicitly state that, it might be useful to the reader to easily interpret setting.
	 As in later iterations of sampling process much of semantics or high-level features are inferred and guidance might not play such a vital role?

**Questions:**

In terms of Suppressed samples illustration and sample diversity, empirical metrics demonstrate on-par or worse diversity except ImageNet 64 x 64? So it is unclear in what setting is proposed method effective and what are current limitations of proposed approach.
            Also, it might be worthwhile to consider evaluation for few interesting sub-sets rather than dataset level, as it might be useful to evaluate in settings  where categories have large sub-classes which needs to capture fine-grained details vs not many sub-classes? Does Pro-G have sensitivity

What is sensitivity of proposed progressive guidance method based on accuracy of noise-aware classifier at different noise-levels? Analyzing such sensitivity and how it effects FID, Precision/Recall w.r.t generation would be informative.
               Also, how challenging is it to train noise-aware classifier? As except few of initial works there aren't many follow up works which use classifier guidance as pointed out by authors on few of challenges. I understand authors used pre-trained checkpoints but it would be informative for community towards better interpretation of applicability of proposed method and classifier guidance more generally.

**Limitations:**

Proposed method is simple and effective and its encouraging to see that classifier guidance is on-par with classifier free guidance in terms of sample quality and Diversity, but the quality boost is marginal.

I am not sure if this is valuable enough contribution in terms of novelty and insights for NeurIPS, as proposed method is straight forward and does not provide extensive novel insights either from Diffusion models behavior or empirical properties of classifier guidance.

---

> ### Author Rebuttal · Authors · 2023-08-09
>
> ## Weakness discussion
>
> ### Weakness 1: Noise-aware classifier performance at different noise-level and the sensitivity associated with $\\gamma$.
>
> In the ablation study in section 6.3, we have already discussed the sensitivity of $\\gamma$ value that has affected the generated image quality. Table 5 in the main paper shows the sensitivity that affects the IS/FID and sFID.
>
> We further follow the suggestion of the reviewer to explore the sensitivity of $\\gamma$ with the performance of noise-aware at different timesteps with image quality as in Table R3:
>
> | $\\gamma$ | FID  | Acc@25 | Acc@75 | Acc@150 | Acc@200 | Acc@250 |
> |----------|------|--------|--------|---------|---------|---------|
> | 0.04     | 5.16 | 00.00  | 0.31   | 20.00   | 78.42   | 100     |
> | 0.06     | 5.4  | 00.00  | 0.31   | 20.31   | 79.06   | 100     |
> | 0.1      | 7.28 | 00.00  | 0.31   | 21.87   | 78.75   | 99.68   |
> | 0.2      | 8.67 | 00.00  | 0.31   | 20.62   | 79.37   | 100     |
>
> Table R3: Sensitivity of $\gamma$ regarding FID and the noise-aware classifier accuracy. Acc@25 means the classifier's accuracy at the $25^{th}$ timestep.
>
> As we can observe, FID is very sensitive with $\gamma$, which means the generated image quality is heavily affected by $\\gamma$. However, the classifier performance at different noise levels has little sensitivity regarding $\gamma$ and has little correlation with the image quality.
>
> ### Weakness 2: Value of $\\gamma$ at different timestep
>
> The value of $\gamma$ does not change following the timesteps. This value represents how fast the information degree should converge to a one-hot vector and is kept constant throughout the sampling process.
>
> ### Weakness 3: Trend of guidance scale value for the primary class
>
> Given $\\gamma = 0.04$, the information degree vector often converges into a one-hot vector at around $50^{th}$ timestep. We can observe the trend in Figure R1 in the attached pdf file in the joint rebuttal. In our experience, guidance plays a significant role in fine-tuning class-specific details at the end of the sampling process.
>
> ## Question discussion
>
> ### Question 1: Clarification on the diversity performance
>
> Diversity improvement is not only available for ImageNet64x64. For each resolution, we have:
>
> **ImageNet64x64**: Figure 4(a), (b) in the main paper
>
> **ImageNet128x128**: Figure 9 (left), (right) in Appendix
>
> **ImageNet256x256**: Table 4 (right Figure) in the main paper, and Figure 10, Figure 16 in Appendix.
>
> All the figures indicate a clear superiority of the proposed ProG compared to vanilla classifier guidance.
>
> *Why do some Recall values in Tables 1 and 3 have similar values between ProG and vanilla guidance?*
>
> Because we keep the same $w$ values as in the original papers. Except ImageNet64x64 ($w=4$) and ImageNet256x256 with unconditional diffusion model ($w=10$), other resolutions such as on ImageNet128, the gradient scale $w = 0.5$, ImageNet256(with EDS) $w= 0.5$, ImageNet256 (conditional diffusion) $w=0.7$. With very small $w$, vanilla guidance can achieve good IS/FID/sFID/Recall value but sacrifice the conditional information, meaning that the generated images do not have class information. Due to this small amount of gradients in the sampling process, the application of ProG in these cases did not bring such significant results in diversity. In this work, we show that, by increasing $w$ to achieve conditional information, ProG helps to avoid sacrificing diversity.
>
> ### Question 2: Limitations of the proposed approach
>
> As discussed in lines 294 to 299 in the main paper, although our proposed ProG successfully alleviates the adversarial effects where the images have high conditional confidence but has many suspicious features, it currently can not solve the case where we achieve low conditional confidence for the image. That is the case where classification information is ignored during the sampling process.
>
>
> ### Question 3: Clarification about the extensive novel insights for the classifier guidance
>
> We INDEED incorporated **7 insights** for the behavior of the vanilla classifier guidance for the diffusion model. (all images after Figure 4 are in Appendix due to the page limitation)
>
> For **diversity suppression**, we have done on the **subset** of many types of breeds in ImageNet:
>
> + [**INS1**] Problem of front-face features collapsing of vanilla classifier guidance in Figure 5 (for Brittany Spaniel), Figure 6 (for English Springer), and Figure 7(for Welsh Springer Spaniel).
> + [**INS2**] Problem of collapsing all images into single pose features in Figure 5 (for Brittany Spaniel)
> + [**INS3**] Problem of green grass background in vanilla guidance for some types of dog Figure 7(for Welsh Sprinter Spaniel)
>
> For **non-robust features construction**, we did mainly on some types of breeds and leopard class of ImageNet  :
>
> + [**INS4**] Problem of losing background in vanilla guidance in Figure 8.
> + [**INS5**] Problem of non-robust features construction of vanilla guidance in Figure 11 (for ImageNet64x64) and Figure 12, 13, 14 (For ImageNet256x256)
>
> For intuition for solving the **two problems**, which are adversarial effects and diversity suppressions:
>
> + [**INS6**] Figure 2 provides the intuition of how the information from other classes helps to avoid the lack of diversity.
> + [**INS7**] Figure 3 provides the intuition on how the information from the tiger can help construct the leopard's robust features.
>
> ### Question 5: The training of the noise-aware classifier
>
> The training of a noise-aware classifier is mostly the same as that of a standard classifier. The only difference is the data augmentation step; in the noise-aware classifier, random noise is added to the image before training. As a result, training a noise-aware classifier is much more straightforward than training a diffusion model. We have tried training the models several times and have had no difficulty during training. The training details and hyperparameters are on page 27 in [10].

---

> > ### Author Response · Authors · 2023-08-12
> > **Looking forward to Reviewer zai6's response**
> >
> > Dear Reviewer zai6,
> >
> > Thank you very much for your thoughtful comments, which help us to have a more comprehensive view of our work. Based on your comments, we have:
> >
> > 1. Provided more analysis about the sensitivity of the proposed ProG with noise-aware classifier performance at different levels. This would be an interesting point of view for the readers since it has not been investigated before.
> > 2. Clarified the hyperparameters problem and provided the trend of the gradient scale given a specific $\gamma$. We believe this is also one of the important aspects that the researchers, who work in this field, should concern about.
> > 3. Clarify that the diversity improvement is not only available for ImageNet64x64, and the larger $w$ is, the more significant improvement we can get.
> > 4. Discussed more about the limitations of the work, as mentioned at the end of the main paper.
> > 5. Provided the insights that we have incorporated in the papers. Some insights have been put in the Appendix due to page limitations.
> > 6. The details of training the noise-aware classifier.
> >
> > We believe that we have addressed all of the concerns of the Reviewer. Please let us know if you still want us to provide more information. We are very happy to solve your concerns.
> >
> > Best regards,
> >
> > Author #3926

---

> > ### Author Response · Authors · 2023-08-17
> > **Looking forward to Reviewer zai6's response**
> >
> > Dear Reviewer zai6,
> >
> > Thank you for your valuable comments on our work. We sincerely hope that our rebuttals have addressed all your concerns. Kindly let us know if you have any other concerns, and we will do our best to address them.
> >
> > Best regards,

---

> > > ### Comment · Reviewer_zai6 · 2023-08-18
> > > **Appreciate Author response**
> > >
> > > I am happy with many of experiments which authors have addressed in rebuttal. By updating the paper with different empirical design choices w.r.t sensitivity of classifier, \gamma and other additional experiments it will be valuable to the community.
> > >
> > > I will increase my rating, overall given the proposed method is on-par with CFG method can authors provide any additional insights on kind of Diffusion Paths or ODE/SDE trajectories which CFG vs Classifier guidance methods take? It does seem that both methods might have different properties.
> > >
> > > I will increase my rating overall and would be great if authors can provide insights on CFG vs Classifier Guidance paths, properties of final models in addition to end-end FID, etc.

---

> > > > ### Author Response · Authors · 2023-08-18
> > > > **Thank you for your support**
> > > >
> > > > Dear Reviewer zai6,
> > > >
> > > > Thank you a lot for your support for our work. We provide the insights for the distribution path of classifier guidance and classifier-free guidance as below:
> > > >
> > > > In [12], the authors have mentioned the distribution shift using guidance. The guidance method helps correct the distribution error accumulated throughout the sampling process. As a result, the path of denoising is corrected to be closer to the path of the diffusion process. That might be true for unsupervised model guidance or classifier guidance.
> > > >
> > > > *How about classifier-free guidance?*
> > > >
> > > > We modified the classifier-free guidance technique a bit to have the same form as the classifier guidance as below:
> > > >
> > > > $\tilde{ \epsilon }_t = (1 + w) \epsilon _{\theta}  (x_t, c) - w \epsilon _{\theta}(x_t) \quad$ (r4)
> > > >
> > > > $\Leftrightarrow  \tilde{\epsilon}_t=\epsilon _{\theta}(x_t, c)   + w(\epsilon _{\theta}(x_t, c) - \epsilon _{\theta}(x_t))\quad$ (r5)
> > > >
> > > > Eq.r4 is the original equation for updating the noise prediction of classifier-free guidance. We will first transform the eq.r4 to the form with classification information as in Eq.2. The term $\epsilon_{\theta}(x_t, c) - \epsilon_{\theta}(x_t)$ can be interpreted as classification information (because we have: **condition diffusion** - **diffusion** = **condition**). We denote this term as $C$. Rewrite the Eq.r5:
> > > >
> > > > $\hat{\epsilon}_t = \epsilon _{\theta}(x_t, c) + w C \quad$ (r6)
> > > >
> > > > Combine Eq.r6 with the DDPM sampling process (refer Eq.(1) in the manuscript) we have:
> > > >
> > > > $$x_{t-1} =  \frac{1}{\sqrt{\alpha_t}} (x_t - \frac{(1- \alpha_t)}{\sqrt{(1 - \bar{\alpha}_t)}} \epsilon _{\theta}) + \underbrace{\frac{(\alpha_t - 1) w}{\sqrt{(1 - \bar{\alpha}_t) \alpha_t}} C} _{classifier \ guidance} + \sigma _t z  \quad (r7)$$
> > > >
> > > > Denote $\mu_t = \frac{1}{\sqrt{\alpha_t}} (x_t - \frac{(1- \alpha_t)}{\sqrt{(1 - \bar{\alpha}_t)}} \epsilon _{\theta})$, we have:
> > > >
> > > > $$x_{t-1} =  \mathcal{N}(\mu_t + \underbrace{\frac{(\alpha_t - 1) w}{\sqrt{(1 - \bar{\alpha}_t) \alpha_t}} C} _{classifier \ guidance}, \sigma _t )  \quad (r8)$$
> > > >
> > > > From Eq.r8, we can see that the term $\frac{(\alpha_t - 1)}{\sqrt{1- \bar{\alpha}_t} \alpha_t}C$ is similar to the term $\sigma^2_t \nabla _{x_t} \log p _{\phi}(y_c|x_t)$ in Equ.2(Manuscript).
> > > >
> > > > Since Eq.r8 and Eq.2 shares the same sampling process with similar interpretation on the classifier guidance. This might indicate that the classifier-free can have the same effects as classifier guidance in correcting the distribution shift due to errors in the denoising process of the diffusion model.
> > > >
> > > > However, the classification term in Eq.r8 has only information from the diffusion model without any additional information from the classifier. This raises a question about whether this scheme is helpful in case of the diffusion model has problems such as overfitting or poorly trained. From this point of view, we believe that classifier guidance which provides additional information, will have better outcomes under specific settings of the diffusion model. This problem is interesting and worth exploring in future work. We thank the reviewer for the thoughtful comment. This seems a very promising line of work in the future to explain techniques such as classifier-free and classifier guidance.
> > > >
> > > > We hope this will solve your question about the Diffusion path between classifier and classifier-free guidance. The main conclusion is that they would help to do the same thing, shifting the distribution closer to the distributions of latent space during diffusion. While both of them can consider similarly in terms of equations, the classifier guidance seems to provide additional information that might be helpful to the sampling process of the diffusion model.
> > > >
> > > > [12] Zhang, Z., Zhao, Z., & Lin, Z. (2022). Unsupervised representation learning from pre-trained diffusion probabilistic models. Advances in Neural Information Processing Systems, 35, 22117-22130.

---

> > > > > ### Comment · Reviewer_zai6 · 2023-08-18
> > > > > **Thanks for response, lets elaborate**
> > > > >
> > > > > I would like to thank authors for reference and comments.
> > > > >
> > > > > I agree and understand that guidance can address distribution shift.
> > > > >
> > > > > To expand my comment on classifier guidance vs classifier-free guidance its more about what the 'Score' would be at different stages of diffusion. From CFG perspective given text or any additional conditioning we are sampling from a conditional distribution where in case of classifier guidance the diffusion model is always sampling for full-data distribution.
> > > > > Given that we are sampling from unconditional distribution, is inference of 'latent' is representational space more challenging in case of classifier guidance compared to CFG and does that have implications in terms of number of sampling steps/discretization of ODE/SDE and paths which the model generate.
> > > > > Though both CFG and Classifier-guidance seem similar from modeling perspective I am not sure if the behavior is equivalent from training/fitting perspective.
> > > > >
> > > > > I understand to get more deeper insights into this might need more extensive analysis towards another work but as authors demonstrate on-par end-end performance, wanted to see if there are any additional insights.
> > > > >
> > > > > Appreciate response. Thanks!

---

### Author Rebuttal · Authors · 2023-08-09

We extend our sincere gratitude to the esteemed reviewers for their insightful and constructive feedback, which has significantly contributed to the enhancement of our manuscript. In this joint reply, we address recurring inquiries raised by multiple reviewers and references for other rebuttals, thereby conserving space and referring to pertinent references to provide comprehensive responses.

*Note: All the references to Table/Figure/Equation that appeared inside the rebuttal will have a mark "r" or "R" before the number to distinguish from numbering in the main paper*

### Gen. Question 1: Extend our proposed ProG for Text-to-Image guidance

In GLIDE, [11] has proposed to extend classifier guidance for text-to-image guidance. The sampling equation for GLIDE is shown below:

$$x_{t-1} = \mu _t + \sigma _t * \mathbf{z} + s \sigma_t^2 \nabla _{x_t} (f(x_t) . g(c))\quad \quad _{(r1)}$$

$f(x_t)$ is the image embedding vector and $g(c)$ is the text or description embedding vector. Equation (r1) is mostly similar to equation (3) in our main paper; the only difference is the gradient term resulted from the similarity between two embedding vectors instead of the classification gradient as in the main paper.

Our proposed ProG is applied to GLIDE in equation (r1) following two scenarios:
1. Given one caption, we will utilize a random set of 1000, 5000, or 10000 captions to act as relevant information during sampling. we have: $$g(c) = \sum_{i=0}^{N+1} s_i g(c_i)\quad\quad_{(r2)}  $$ with $i = 0$ is the index of the primary caption, and $i \neq 0$ represents other captions. The initial  values of $s_i$ are set as: $$s_i = \frac{g(c_0) . g(c_i)}{\sum^{N+1}_{j=0} g(c_0) .g(c_j)} \quad \quad _{(r3)}$$ The value of $s_i$ is progressively updated during sampling as in section 3.2 in the main paper. This scheme is named **GLIDE-ProG**
2. Given one caption, we use four other captions that have the same meaning as the original caption but different words. Since four other captions, all have the same meaning; we have different strategies to set the $s_i$ values: $$s_i = \begin{cases}a, \quad \text{if } i = 0\\\ \frac{1-a}{4}, \quad \text{otherwise} \end{cases}$$, with $0 \leq a \leq 1$ is hyperparameter, we try with $a=0.3$. This method is named **GLIDE-ProGsim**

We set up an evaluation like GLIDE [11] to evaluate zero-shot FID on MS-CoCo. Note: 4 additional equivalent captions of GLIDE-ProGsim are taken from the set of captions available for each image in MS-Coco captions. 1k, 5k, and 10k captions are randomly sampled from the MSCoco training set. Table R1 and Table R2 show the evaluation results:

| | zero-shot FID | computational cost (GPU hours)|
|:----:|:------:|--------|
| GLIDE |    24.80  | 34.27  |
| GLIDE-ProG w N=1k |  23.47| 34.66 |
| GLIDE-ProG w N=5k |23.50|	34.83 |
| GLIDE-ProG w N=10k|	 **23.31**|34.83|
| GLIDE-ProGsim | 23.87 |34.84 |

Table R1: MSCoCo64x64 zero-shot FID evaluation where 30000 captions are sampled from the MSCoco validation set.

|| zero-shot FID | computational cost (GPU hours) |
|:---:|:--:|---|
| GLIDE  | 34.80| 38.45  |
| GLIDE-ProG w N=1k | 32.55 |    45.50 |
| GLIDE-ProG w N=5k |32.37	|45.80|
| GLIDE-ProG w N=10k|	32.28|46.10|
| GLIDE-ProGsim 	|  **31.91**  |46.23|

Table R2: MSCoCo256x256 zero-shot FID evaluation where 30000 captions are sampled from the MSCoco validation set.


**Conclusion**: From Table R1 and Table R2, the ProG scheme helps significantly improve the performance of text-to-image guidance in different scenarios with low additional computational costs. Given the many captions available, we can use the first scenario to improve the generated images. Otherwise, the second scenario is also very easy to implement. The additional captions can be gathered from Large Language Models (LLMs) to generate images.

**Reference**:

[1] Ho, J., & Salimans, T. (2022). Classifier-free diffusion guidance. arXiv preprint arXiv:2207.12598.

[2] Gao, S., Zhou, P., Cheng, M. M., & Yan, S. (2023). A masked diffusion transformer is a strong image synthesizer. arXiv preprint arXiv:2303.14389.

[3]  Kim, D., Kim, Y., Kwon, S.J., Kang, W. &amp; Moon, I.. (2023). Refining Generative Process with Discriminator Guidance in Score-based Diffusion Models. Proceedings of the 40th International Conference on Machine Learning

[4] Hang, T., Gu, S., Li, C., Bao, J., Chen, D., Hu, H., ... & Guo, B. (2023). Efficient diffusion training via min-snr weighting strategy. arXiv preprint arXiv:2303.09556. (ICCV2023)

[5] Peebles, W., & Xie, S. (2022). Scalable diffusion models with transformers. arXiv preprint arXiv:2212.09748.

[6] Sauer, A., Schwarz, K., & Geiger, A. (2022, July). Stylegan-xl: Scaling stylegan to large, diverse datasets. In ACM SIGGRAPH 2022 conference proceedings (pp. 1-10).

[7] Singh, R., Shukla, A., & Turaga, P. (2023). Polynomial Implicit Neural Representations For Large Diverse Datasets. In Proceedings of the IEEE/CVF Conference on Computer Vision and Pattern Recognition (pp. 2041-2051).

[8] Ganz, Roy, and Michael Elad. "BIGRoC: Boosting Image Generation via a Robust Classifier." Transactions on Machine Learning Research (2023).

[9] Dinh, A., Liu, D. &amp; Xu, C.. (2023). PixelAsParam: A Gradient View on Diffusion Sampling with Guidance. Proceedings of the 40th International Conference on Machine Learning

[10] Dhariwal, P., & Nichol, A. (2021). Diffusion models beat gans on image synthesis. Advances in neural information processing systems, 34, 8780-8794.

[11]
Nichol, A.Q., Dhariwal, P., Ramesh, A., Shyam, P., Mishkin, P., Mcgrew, B., Sutskever, I. &amp; Chen, M.. (2022). GLIDE: Towards Photorealistic Image Generation and Editing with Text-Guided Diffusion Models.Proceedings of the 39th International Conference on Machine Learning

---

> ### Author Response · Authors · 2023-08-15
> **Clarify the information in the Table R1 and R2**
>
> Dear Reviewers,
>
> Thanks for your valuable review. I am clarifying the information in Table R1 and R2.
>
> In Tables R1 and R2, all the methods **GLIDE-ProG w N=1k**, **GLIDE-ProG w N=5k**, **GLIDE-ProG w N=10k** and **GLIDE-ProGsim** are our proposed methods which is the combination between our proposed **ProG** with GLIDE to improve the text-to-image guidance with different scenarios as described above.
>
> |               	| zero-shot FID ($\downarrow$) | computational cost (GPU hours)|
> |:----------------:|:-------------:|--------------------|
> | GLIDE         	|    24.80      |       34.27          |
> | **GLIDE-ProG w N=1k (ours)** |    23.47      |     34.66          |
> | **GLIDE-ProG w N=5k (ours)** |	 23.50		|		      34.83			     |
> | **GLIDE-ProG w N=10k (ours)**|	 **23.31**		|			34.83		       |
> | **GLIDE-ProGsim (ours)**	|    23.87      |       34.84          |
>
> Table R1: The improvement is significant in all the scenarios (around **6%**), with N is the number of additional captions we used. Dataset: MSCoco64x64
>
>
> |               	| zero-shot FID ($\downarrow$)  | computational cost (GPU hours) |
> |:----------------:|:-------------:|--------------------|
> | GLIDE         	|      34.80    |      38.45              |
> |**GLIDE-ProG w N=1k (ours)** |      32.55    |    45.50                |
> | **GLIDE-ProG w N=5k (ours)** |	   32.37	|				45.80	 |
> | **GLIDE-ProG w N=10k (ours)**|	   32.28	|				46.10	 |
> | **GLIDE-ProGsim (ours)** 	|      **31.91**    |   46.23                 |
>
> Table R2: The improvement is significant in all the scenarios (around **8%**), with N is the number of additional captions we used. Dataset: MSCoco256x256.
>
> If you still have any concerns, please let us know. We will do our best to address them.
>
> Best regards,
>
> Author #3926

---

### Decision · Program_Chairs · 2023-09-21

**Decision:**

Accept (poster)

**Comment:**

The paper received all positive reviews. Some minor technical concerns about writing and experiments raised by the reviewers were then solved by the rebuttal. Overall this work points out that previous diffusion models with classifier guidance only focus on the given category while ignoring the other relevant category information, and proposes the Progressive Guidance (PG) method to address the problems, allowing relevant classes' gradients to contribute to shared information construction.  Such method can be of large interest to the community working on classifier-guided diffusion models.